# FMRP promotes RNA localization to neuronal projections through interactions between its RGG domain and G-quadruplex RNA sequences

Raeann Goering[1], Laura I Hudish[2], Bryan B Guzman[3], Nisha Raj[4], Gary J Bassell[4], Holger A Russ[2], Daniel Dominguez[3], J Matthew Taliaferro[1,5]*

[1]Department of Biochemistry and Molecular Genetics, University of Colorado Anschutz Medical Campus, Boulder, United States; [2]Barbara Davis Center for Diabetes, University of Colorado Anschutz Medical Campus, Boulder, United States; [3]Department of Pharmacology, University of North Carolina at Chapel Hill, Chapel Hill, United States; [4]Departments of Cell Biology and Neurology, Emory University School of Medicine, Atlanta, Georgia; [5]RNA Bioscience Initiative, University of Colorado Anschutz Medical Campus, Boulder, United States

**Abstract** The sorting of RNA molecules to subcellular locations facilitates the activity of spatially restricted processes. We have analyzed subcellular transcriptomes of FMRP-null mouse neuronal cells to identify transcripts that depend on FMRP for efficient transport to neurites. We found that these transcripts contain an enrichment of G-quadruplex sequences in their 3′ UTRs, suggesting that FMRP recognizes them to promote RNA localization. We observed similar results in neurons derived from Fragile X Syndrome patients. We identified the RGG domain of FMRP as important for binding G-quadruplexes and the transport of G-quadruplex-containing transcripts. Finally, we found that the translation and localization targets of FMRP were distinct and that an FMRP mutant that is unable to bind ribosomes still promoted localization of G-quadruplex-containing messages. This suggests that these two regulatory modes of FMRP may be functionally separated. These results provide a framework for the elucidation of similar mechanisms governed by other RNA-binding proteins.

*For correspondence:
matthew.taliaferro@cuanschutz.edu

**Competing interests:** The authors declare that no competing interests exist.

## Introduction

Essentially all eukaryotic cells contain subcellular regions that are associated with specific functions. The identity and activity of these regions are in large part defined by the proteins contained within them. Cells require an efficient sorting process to get proteins required for function to their correct location. For many genes, this fundamental problem is solved by transporting the RNA molecule encoding the protein, rather than the protein itself, to the desired location. The transcript is then translated on-site to immediately produce a correctly localized protein.

RNA localization has been studied extensively in diverse systems (*Ephrussi et al., 1991*; *Long, 1997*; *Moor et al., 2017*). Thousands of transcripts are asymmetrically distributed within cells and therefore subject to localization regulation (*Cajigas et al., 2012*; *Gumy et al., 2011*; *Lécuyer et al., 2007*; *Taliaferro et al., 2016a*; *Taylor et al., 2009*). However, precise molecular mechanisms that govern how transcripts are transported have been worked out in detail for only a small subset of heavily studied localized RNAs (*Bertrand et al., 1998*; *Ephrussi et al., 1991*; *Jambhekar and Derisi, 2007*; *Kislauskis et al., 1994*). In these RNAs, sequences that mark transcripts for transport, sometimes termed 'zipcodes', are often found in 3′ UTRs (*Jambhekar and*

*Derisi, 2007*). Zipcode sequences are then bound by RNA-binding proteins (RBPs) that mediate RNA transport (*Farina et al., 2003*; *Huang et al., 2003*; *St Johnston et al., 1991*; *Zhang et al., 2001*). The zipcode/RBP pairs that are known were often identified through careful but laborious reporter experiments that involve systematic deletions or mutations that inactivate putative regulatory elements.

A large disparity persists, though, between the number of RNAs that are targets of localization regulation and the number of known localization regulatory elements within those RNAs. High-throughput experiments that probe the localization of thousands of transcripts at once may be able to help bridge this gap. By providing many examples of transcripts whose localization is regulated by a given RBP, commonalities among those transcripts can shed light on general rules that define sequence features that regulate RNA localization as well as the RBP that binds them.

FMRP is an RBP that is highly expressed in the brain and has long been known to be the protein behind Fragile X syndrome (FXS) (*Santoro et al., 2012*). Loss of expression of the *FMR1* gene in humans is associated with intellectual disabilities and occurs in approximately 1 in 5000 males (*Coffee et al., 2009*). FMRP-null mice display similar phenotypes (*Kazdoba et al., 2014*). FMRP has been shown to regulate RNA metabolism at the level of translational repression and RNA localization (*Darnell et al., 2011*; *Dictenberg et al., 2008*). The relative contribution of these activities to observed phenotypes is generally unclear. Although genome-wide studies probing the translation repression activity of FMRP have been performed (*Darnell et al., 2011*), much less is known about the RNAs that depend on FMRP for efficient localization to neuronal projections. FMRP has been implicated in the transport of a handful of RNAs (*Dictenberg et al., 2008*; *Muddashetty et al., 2007*; *Zalfa et al., 2007*), but there is no general understanding of the identity of transcripts whose localization is regulated by FMRP.

How FMRP recognizes and binds its functional targets is also not well understood. Several studies have reported transcript and RNA motif targets for FMRP in vitro and in vivo (*Ascano et al., 2012*; *Brown et al., 2001*; *Darnell et al., 2001*, *Darnell et al., 2005a*; *Vasilyev et al., 2015*), yet considerable disagreement among these studies remains about the identity of RNA sequences that interact with FMRP with high-affinity (*Anderson et al., 2016*). RNA G-quadruplex sequences have been repeatedly identified as binding targets of FMRP (*Blice-Baum and Mihailescu, 2014*; *Darnell et al., 2001*; *Schaeffer et al., 2001*; *Zhang et al., 2014*), and studies that have looked at commonalities between published datasets have found enrichments for WGGA sequences consistent with G-quadruplex formation (*Anderson et al., 2016*; *Suhl et al., 2014*). However, other studies have found that FMRP binds completely different short RNA motifs (*Ray et al., 2013*) while still others seem to show that FMRP displays almost no sequence specificity at all and instead binds all along the coding sequence of its target transcripts (*Darnell et al., 2011*). Whether or not these potentially different modes of binding are related to the different functions of FMRP is again unknown.

## Results

### Identification of transcripts that depend upon FMRP for efficient neuronal transport

We and others have previously used subcellular fractionation followed by high-throughput sequencing to quantify transcripts in the soma and neurites of neuronal cells (*Taliaferro et al., 2016a*; *Zappulo et al., 2017*). In these experiments, neuronal cells are plated on porous membranes (*Figure 1A*). The pores of these membranes are big enough to allow neurite growth through them but small enough to restrict soma growth to the top of the membrane. This allows the cells to be mechanically fractionated into soma and neurite fractions. RNA can then be isolated from both fractions and analyzed by high-throughput sequencing.

Given the utility of this technique, we then set out to define transcripts whose efficient localization to neurites depended on FMRP. Using CRISPR/Cas9, we created an FMRP-null CAD mouse neuronal cell line. CAD cells are derived from a mouse brain tumor and can be induced by serum starvation to display features of neuronal differentiation, including cell cycle downregulation, expression of neuronal markers, and the appearance of neuronal morphologies (*Qi et al., 1997*). The CAD FMRP knockout line was found to have a single basepair deletion in all alleles of *Fmr1* (*Figure 1—figure supplement 1*) that caused a frameshift and a premature termination codon two codons

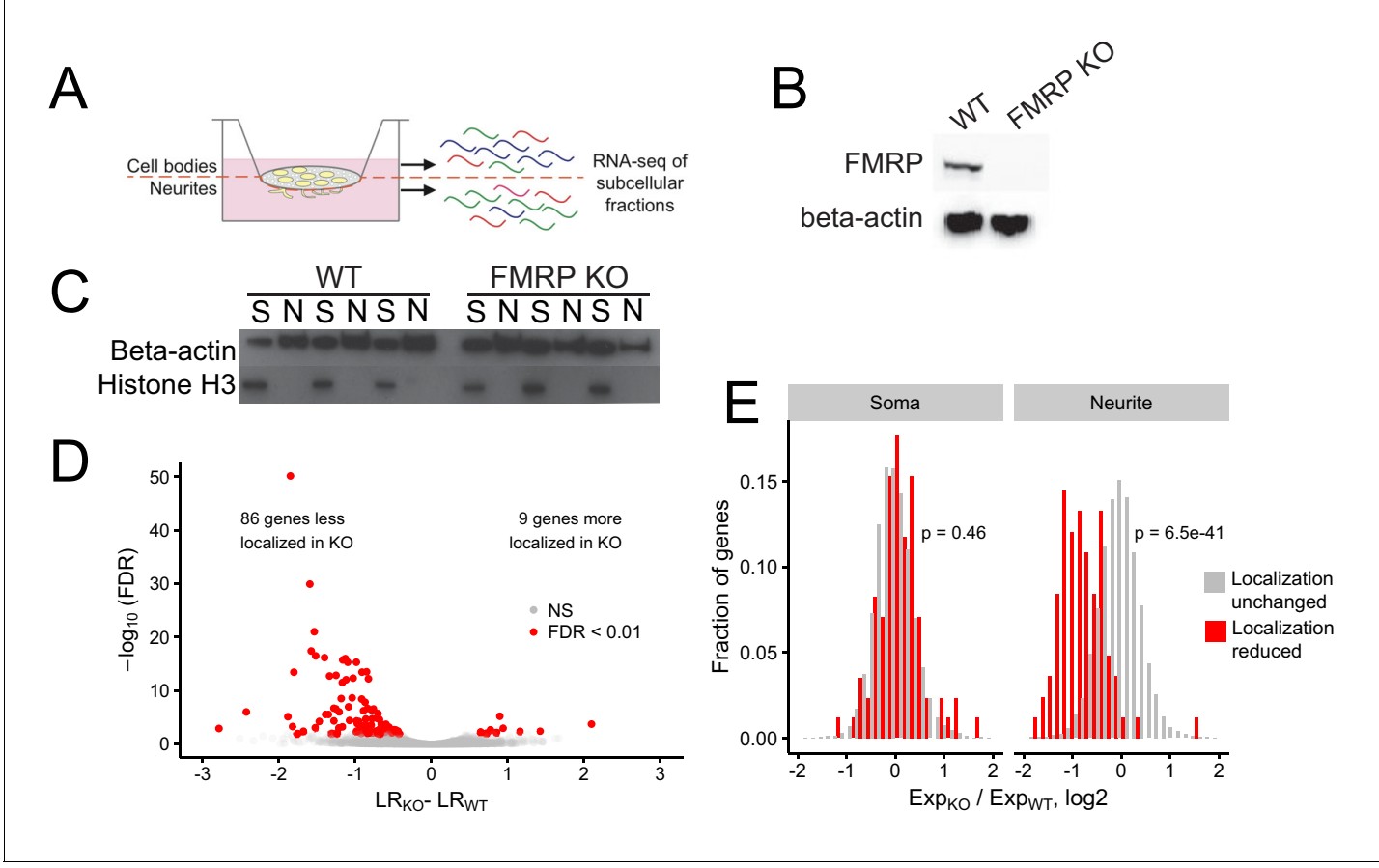

**Figure 1.** Identification of FMRP localization targets. (**A**) Schematic of soma/neurite fractionation. Cells are plated on porous membranes. Neurites grow down through the pores, and cells are then mechanically fractionated by scraping. (**B**) Western blot of wildtype and CAD FMRP knockout cells. (**C**) Western blot of fractionated soma (S) and neurite (N) samples. Beta-actin is a marker of both soma and neurite fractions while Histone H3, being restricted to the nucleus, is a marker of soma fractions. (**D**) Localization ratio (LR) comparison for all expressed genes in wildtype and FMRP-null cells. Changes in LR values and FDR values were calculated using Xtail. (**E**) Expression changes across genotypes in the soma and neurite fractions for FMRP localization targets (red) and nontargets (gray). P-values were calculated using a Wilcoxon rank-sum test.

The online version of this article includes the following figure supplement(s) for figure 1:

**Figure supplement 1.** Location of guide RNA directed against *Fmr1*.
**Figure supplement 2.** RNA expression levels of *Fmr1* in the wildtype and FMRP null cells.
**Figure supplement 3.** PCA analysis of gene expression values from the soma and neurite compartments of wildtype and FMRP null cells.
**Figure supplement 4.** Correlation and clustering analysis of gene expression values from the soma and neurite compartments of wildtype and FMRP null cells.
**Figure supplement 5.** LR values for all genes (gray), ribosomal protein genes (red), and genes that are part of the electron transport chain (blue).
**Figure supplement 6.** Clustering of LR values from 65 neuronal subcellular transcriptomic experiments.
**Figure supplement 7.** Standard deviation of LR values for each gene across all samples.
**Figure supplement 8.** LR values for genes encoding ribosomal proteins for all samples.
**Figure supplement 9.** LR values for genes encoding members of the electron transport chain for all samples.
**Figure supplement 10.** LR values for genes in wildtype and FMRP null cells.
**Figure supplement 11.** Gene ontology enrichments for genes with reduced neurite localization in FMRP knockout CAD cells.

downstream of the gRNA cut site. The premature termination codon was upstream of all identified RNA-binding domains in FMRP. The knockout cells displayed no detectable FMRP expression by immunoblotting, even when probed with a polyclonal FMRP antibody (*Figure 1B*).

We then fractionated wildtype and FMRP knockout CAD cells into soma and neurite fractions in triplicate. To monitor the efficiency of the fractionation, we assayed the expression of beta-actin, which should be present in all fractions, and histone H3, which, being nuclear, should be restricted

to the soma (*Figure 1C*). RNA was isolated from each fractionation and profiled by high-throughput sequencing.

The expression of *Fmr1* RNA was downregulated in FMRP knockout samples, as would be expected since the Cas9-induced lesion likely made the RNA a target of nonsense-mediated decay (*Figure 1—figure supplement 2*). Principal component analysis of gene expression values showed that replicates clustered together and samples were separated along the first two principal components by compartment and genotype, respectively (*Figure 1—figure supplements 3* and *4*). To quantify RNA localization, we defined a metric termed localization ratio (LR) as the log2 of a gene's abundance in the neurite fraction divided by its abundance in the soma fraction. Positive LR values are therefore associated with neurite localization, and higher LR values are associated with stronger neurite localization.

Previous literature has demonstrated that transcripts encoding ribosomal proteins and components of the electron transport chain are strongly enriched in neurites (*Gumy et al., 2011*; *Moccia et al., 2003*; *Taliaferro et al., 2016a*). We therefore sought to assay the efficiency of our fractionation and RNA isolation by looking at the LR values of these genes. Reassuringly, we found that, in all samples, mRNAs encoding these proteins were strongly neurite-enriched (*Figure 1—figure supplement 5*).

To further monitor the quality of our fractionations and RNA samples, we compared data produced by these fractionations to data produced by similar neuronal subcellular transcriptomic experiments that had been previously published. In total, we calculated LR values for all expressed genes from 65 samples spanning two species, four laboratories, and nine neuronal cell types (*Farris et al., 2019*; *Minis et al., 2014*; *Taliaferro et al., 2016a*; *Zappulo et al., 2017*; *Hudish et al., 2020 Figure 1—figure supplement 6*). We found that our LR values correlated well with LR values from other studies (R ~ 0.6 to 0.8) and that all 65 samples were significantly more similar to each other than would be expected by chance (*Figure 1—figure supplement 7*), giving us confidence that our subcellular sequencing data was comparable with previously published datasets. Furthermore, we found that transcripts encoding ribosomal proteins (*Figure 1—figure supplement 8*) and components of the electron transport chain (*Figure 1—figure supplement 9*) were neurite-enriched in a large majority of samples, confirming their status as markers of neurite RNA.

To identify genes whose transcripts depend on FMRP for efficient transport to neurites, we compared the LR values of genes between the wildtype and knockout samples using the R package Xtail (*Xiao et al., 2016*). Xtail is designed to analyze ribosome profiling data to identify genes whose translational efficiency changes across conditions. However, since both translational efficiency and LR are ratios of gene expression values, we reasoned that Xtail would be a good tool to identify genes whose LR value changes across samples. Using Xtail and a false discovery rate cutoff of 0.01, we identified 86 genes whose transcripts displayed impaired neurite localization, and thus decreased LR values, in the FMR1 knockout (*Figure 1D*, *Figure 1—figure supplement 10*, *Supplementary file 1a*). We defined these genes as FMRP localization targets.

Gene ontology analysis of these targets revealed that they were enriched for genes that regulate vesicle transport in neurons (*Figure 1—figure supplement 11*). In particular, mRNAs encoding several kinesins were less localized in the FMRP knockout. Interestingly, mRNAs encoding kinesins have been recently identified as both bound and transported by FMRP in glial cells (*Pilaz et al., 2016*).

Because LR is a ratio of neurite to soma gene expression values, in principle, a decrease in LR could be due to either a decrease in neurite expression or an increase in soma expression. To investigate this further, we compared the expression of the mislocalized genes in the soma and neurite compartments separately. We found that while the expression of FMRP localization targets did not change in the soma between the wildtype and FMRP knockout samples, their expression in the neurites of FMRP knockout cells was significantly reduced (*Figure 1E*). Because the soma contains the vast majority of cellular RNA, this indicates that the overall expression and processing of the localization targets is not impaired yet their ability to be trafficked to neurites is reduced. These findings suggest that these genes are in fact misregulated at the level of RNA localization.

## FMRP localization targets have significant enrichments of G-quadruplex sequences in their 3′ UTRs

After identifying functional RNA localization targets of FMRP, we then looked for commonalities among them that can give insight into how FMRP recognizes and binds them. If FMRP were directly

regulating the localization of these targets, we would expect that FMRP directly binds them. To test this, we analyzed data from three previously published FMRP CLIP-seq datasets (*Ascano et al., 2012*; *Darnell et al., 2011*; *Maurin et al., 2018*). Given the lack of consensus between multiple published FMRP binding datasets (*Anderson et al., 2016*), we defined transcripts bound by FMRP as those that appeared in all three CLIP-seq datasets (*Ascano et al., 2012*; *Darnell et al., 2011*; *Maurin et al., 2018*). FMRP localization targets were approximately 4-fold more likely to be directly bound by FMRP than expected by chance (*Figure 2A*).

We then performed the converse analysis to ask whether FMRP CLIP-seq targets were preferentially mislocalized in FMRP knockout cells. We found that the localization of FMRP CLIP-seq targets was significantly more sensitive to FMRP loss than CLIP-seq nontargets (*Figure 2B*).

To rigorously test the links between these three published CLIP-seq datasets and our FMRP localization targets, we then repeated these analyses using the FMRP CLIP-seq targets defined in each dataset individually instead of taking the overlap between them. For each CLIP-seq dataset, we found that our FMRP localization targets were enriched for FMRP binding (*Figure 2—figure supplements 1*, *3* and *5*) and that FMRP-bound RNAs were significantly more likely to be mislocalized in FMRP knockout cells (*Figure 2—figure supplements 2*, *4* and *6*). From these analyses, we conclude that FMRP is likely directly regulating the localization of these functional RNA localization targets.

We then reasoned that sequences that were enriched in FMRP localization targets relative to nontargets could represent preferred binding sites for FMRP. Because the majority of RNA sequence elements that regulate localization identified to date reside in 3′ UTRs (*Jambhekar and Derisi, 2007*), we compared the abundance of all 6mers in the 3′ UTRs of FMRP localization targets and nontargets. We observed a strong enrichment for the RNA sequence 'GGA' in the 3′ UTR of FMRP localization targets (*Figure 2C*). GGA had previously been identified from an FMRP CLIP-seq dataset as a preferred FMRP binding motif, giving us further confidence that our identified localization targets were directly bound and regulated by FMRP (*Ascano et al., 2012*).

Tandem repeats of GGA sequences are known to have the ability to fold into G-quadruplex sequences (*Suhl et al., 2014*). G-quadruplexes are four-stranded nucleic acid structures in which guanosine residues interact with each other across the strands (*Rhodes and Lipps, 2015*). Given that G-quadruplex structures have been shown to be specifically bound by FMRP in vitro, (*Didiot et al., 2008*; *Vasilyev et al., 2015*), we then asked if G-quadruplex sequences were enriched in FMRP localization target sequences. To do so, we employed two computational methods to predict G-quadruplex sequences from RNA sequences. Using a regular expression we searched for four repeated WGGA sequences separated by 0–7 nucleotide linkers (*Huppert and Balasubramanian, 2007*) where W represents either adenosine or uracil. We also used the RNA secondary structure prediction software RNAfold (*Lorenz et al., 2011*). Both methods agreed that the 3′ UTRs of FMRP localization targets contained significantly more G-quadruplex sequences than nontargets (*Figure 2D,E*).

We then complemented these computational predictions of G-quadruplex sequences with previously published experimentally defined G-quadruplex sequences (*Guo and Bartel, 2016*). These were identified by quantifying reverse transcriptase stop sites in the presence of either potassium or lithium. Because potassium stabilizes G-quadruplexes and lithium destabilizes them, stop sites that display differential abundance between the two conditions are likely due to G-quadruplexes. These experimentally defined G-quadruplexes displayed many expected properties, including a large enrichment for guanosine nucleotides (*Figure 2—figure supplement 7*), highly stable predicted secondary structures (*Figure 2—figure supplement 8*), and a strong enrichment for predicted G-quadruplexes using both the regular expression and RNAfold methods (*Figure 2—figure supplements 9* and *10*). The agreement between the computational and experimental approaches to G-quadruplex identification gave us confidence in the ability of both techniques. As with the computational predictions, we observed a significant enrichment of experimentally defined G-quadruplexes in the 3′ UTRs of FMRP localization targets (*Figure 2F*).

We then asked what other transcript features were associated with FMRP localization targets. We observed that FMRP localization targets had much longer open reading frames and 3′ UTRs than expected (*Figure 2G*), consistent with previous reports that 3′ UTR extensions and alternative 3′ UTRs can regulate RNA localization in neurons (*Taliaferro et al., 2016a*; *Tushev et al., 2018*). We observed no relationship between the overall abundance of a transcript and whether or not it was an FMRP localization target. The enrichment of G-quadruplexes in FMRP localization targets was

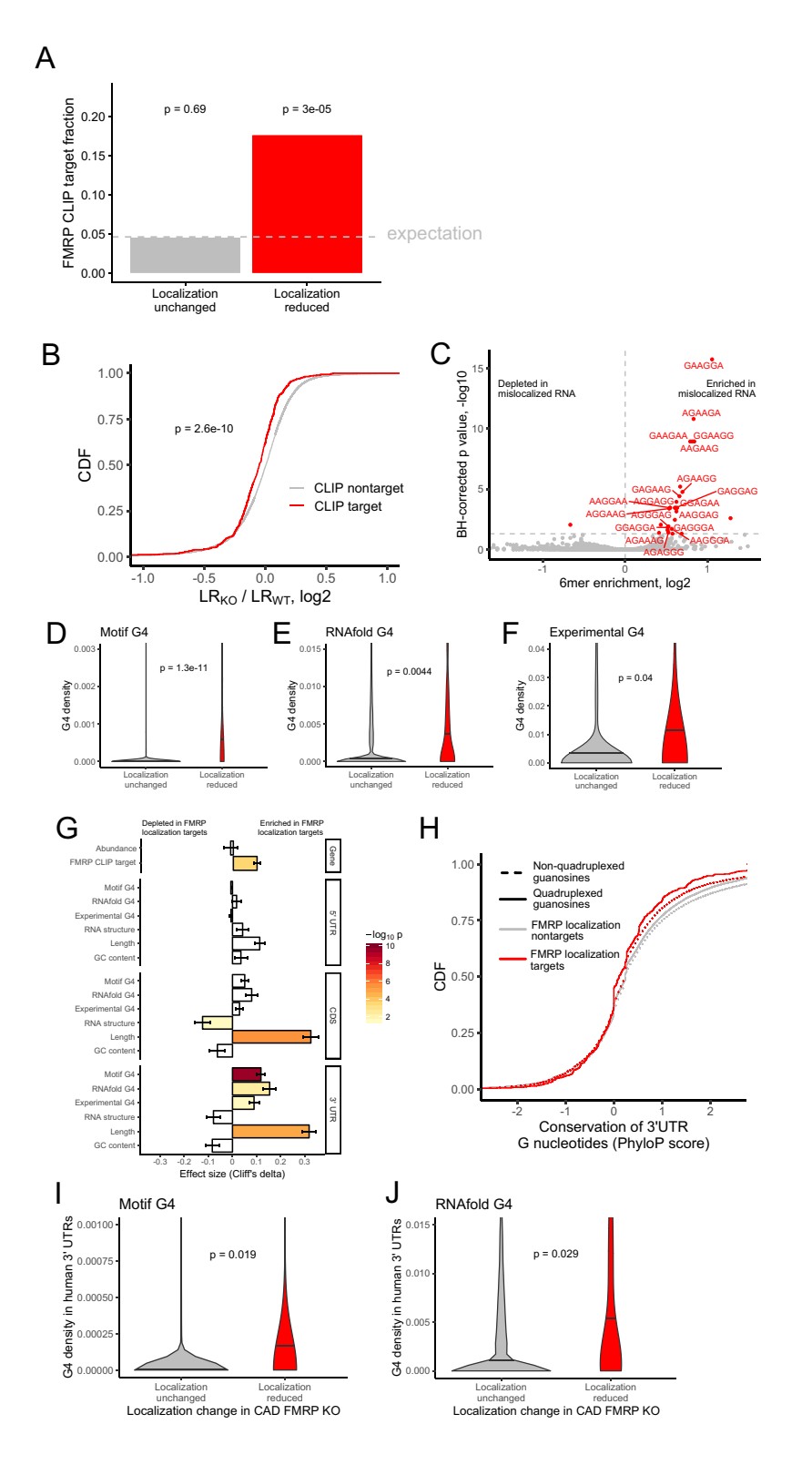

**Figure 2.** Identification of transcript features associated with the regulation of localization by FMRP. (**A**) Fraction of FMRP localization targets (red) and nontargets (gray) that were previously identified as directly bound by FMRP in cells. FMRP-bound RNAs were defined as the intersection between three previously published datasets (*Ascano et al., 2012*; *Darnell et al., 2011*; *Maurin et al., 2018*). The null, expected fraction is represented by a gray dotted line. P values were calculated using a binomial test. (**B**) Change in RNA localization (knockout - wildtype) for RNAs identified as CLIP targets

*Figure 2 continued on next page*

*Figure 2 continued*

(red) and CLIP nontargets (gray). P values were calculated using a Wilcoxon rank-sum test. (C) Sixmer enrichments in the 3′ UTRs of FMRP localization targets vs nontargets. Significantly enriched 6mers (BH-adjusted p<0.05, Fisher's exact test) are represented in red. (D-F) G-quadruplex sequence densities in the 3′ UTRs of FMRP localization targets (red) and nontargets (gray). P values are calculated using a Wilcoxon rank-sum test. In (D), G-quadruplex sequences were defined by a regular expression that contained four WGGA sequences separated by linkers of length 0–7 nt. Values represent sequence matches per nt. In (E), RNAfold was used to identify G-quadruplex sequences. Values represent the number of guanosine residues participating in quadruplex sequences per nt. In (F), experimentally defined G-quadruplex sequences were used. Values represent the number of basepairs that overlap with an experimentally defined G-quadruplex sequence per nt of UTR sequence. (G) Effect sizes and significance of the relationships between transcript features and whether or not a transcript's localization was regulated by FMRP. (H) Conservation of guanosine residues in 3′ UTRs as defined by PhyloP scores. Guanosines were separated based on whether or not they were predicted by RNAfold to participate in quadruplexes and by whether or not they were contained in FMRP localization targets. (I) G-quadruplex sequence density in the 3′ UTRs of the human orthologs of FMRP targets. G-quadruplexes were defined by the regular expression used in (D), and values represent quadruplex sequences per nt. (J) G-quadruplex sequence density in the 3′ UTRs of the human orthologs of FMRP targets. G-quadruplex sequences were defined by RNAfold as in (E), and values represent the number of guanosine residues participating in quadruplex sequences per nt.

The online version of this article includes the following figure supplement(s) for figure 2:

**Figure supplement 1.** Overlap between FMRP functional localization targets and FMRP CLIP-seq targets as identified in Ascano et al.

**Figure supplement 2.** Change in RNA localization in FMRP knockout cells compared to wildtype for FMRP CLIP-seq targets (red) and non-targets (gray) as identified by Ascano et al.

**Figure supplement 3.** Overlap between FMRP functional localization targets and FMRP CLIP-seq targets as identified in Darnell et al.

**Figure supplement 4.** Change in RNA localization in FMRP knockout cells compared to wildtype for FMRP CLIP-seq targets (red) and non-targets (gray) as identified by Darnell et al.

**Figure supplement 5.** Overlap between FMRP functional localization targets and FMRP CLIP-seq targets as identified in Maurin et al.

**Figure supplement 6.** Change in RNA localization in FMRP knockout cells compared to wildtype for FMRP CLIP-seq targets (red) and non-targets (gray) as identified by Maurin et al.

**Figure supplement 7.** Experimentally-defined G-quadruplex sequences contain more guanosine residues than control sequences drawn from the same genes.

**Figure supplement 8.** Minimum free energy structures calculated using RNAfold show the G-quadruplex structures are significantly more stable than control sequences drawn from the same genes.

**Figure supplement 9.** Experimentally defined G-quadruplex sequences contain many more regular expression-defined G-quadruplex sequences than controls.

**Figure supplement 10.** Experimentally defined G-quadruplex sequences contain many more RNAfold-defined quadruplexed guanosine residues than control sequences.

restricted to their 3′ UTRs as we did not observe a significant relationship between CDS or 5′ UTR G-quadruplex density and localization regulation by FMRP (*Figure 2G*).

## G-quadruplexes are not positionally conserved but tend to be present in orthologous genes

The enrichment of G-quadruplex sequences in FMRP localization targets and the enrichment of those targets in FMRP CLIP-seq datasets suggested that the G-quadruplex sequences may be functional. If they were functional, we might expect them to be conserved. To test the conservation of G-quadruplex sequences, we compared the conservation of 3′ UTR guanosine residues predicted by RNAfold to participate in quadruplex interactions to those not predicted to be in quadruplexes. Surprisingly, we found that guanosines in quadruplexes were less conserved than those not in quadruplexes, and those in FMRP localization targets were less conserved than those in nontargets (*Figure 2H*). However, this analysis relies on elements being positionally conserved such that they can be aligned across genomes. Elements within UTRs do not have to be positionally conserved, only present, to be functional.

To assess this, we calculated G-quadruplex densities in the 3′ UTRs of the human orthologs of the FMRP localization targets and nontargets. We observed a significant enrichment of G-quadruplexes in the human orthologs of the FMRP localization targets, arguing that their function may be conserved across species (*Figure 2I,J*).

## Mislocalized transcripts in FXS patient-derived neurons are enriched for G-quadruplex sequences

To more directly test the localization regulatory activity of FMRP in human cells, we created induced pluripotent stem (iPS) cell lines from FXS and unaffected patient samples. We differentiated these cells over the course of 20 days into motor neurons (*Chambers et al., 2009*; *Du et al., 2015*; *Reinhardt et al., 2013*; *Hudish et al., 2020*; *Figure 3—figure supplement 1*). We then plated these cells on porous membranes and performed the same subcellular fractionation and sequencing experiment that was performed on the CAD cell samples using four technical replicates.

As expected, the FXS neurons displayed a large decrease in both *FMR1* RNA and FMRP protein (*Figure 3A,B*). We assayed the efficiency of the fractionation as before by immunoblotting (*Figure 3—figure supplement 2*) and looking for the enrichment of mRNAs encoding ribosomal proteins and components of the electron transport chain in the neurite fraction (*Figure 3—figure supplement 3*) and found that both performed as expected. Further, LR values obtained from the iPS neurons correlated well (R ~ 0.6 to 0.8) with those obtained from previously published neuronal subcellular sequencing datasets and with the CAD cell fractionation in *Figure 1* (*Farris et al., 2019*; *Minis et al., 2014*; *Taliaferro et al., 2016a*; *Zappulo et al., 2017*; *Hudish et al., 2020*; *Figure 1— figure supplement 6*), indicating that the fractionation was efficient.

PCA analysis of gene expression values showed a separation of samples by compartment and genotype along the first two principal components (*Figure 3—figure supplement 4*). We first looked at the localization of the human orthologs of the FMRP localization targets that were identified in the mouse CAD system. Strikingly, the human orthologs of the mouse FMRP localization targets were significantly less neurite-localized in FXS neurons than in unaffected neurons, indicating that the functional localization targets of FMRP were broadly conserved across species (mouse and human) and experimental systems (neuronal cell line and iPS-derived neurons) (*Figure 3C*). Overall, we observed a modest (R = 0.21) but highly significant (p=1e-37) correlation between the mislocalization of genes in the CAD knockout system and the mislocalization of their human orthologs in the FXS iPS system (*Figure 3—figure supplement 5*, *Supplementary file 1b*).

We then defined genes whose localization differed between the FXS and unaffected samples. After quantifying the localization of genes in each sample, we found that very few genes met a reasonable FDR cutoff for mislocalization. This is perhaps due to the inherent variability in the differentiation of iPS cells into motor neurons from prep to prep or the difference in genetic background of the iPS lines adding noise to gene expression measurements (*Figure 3—figure supplement 4*). To define mislocalized genes, we therefore used a log2 fold change cutoff of 0.25 to define genes whose neurite-localization decreased in FXS (FMRP targets) or increased in FXS (FMRP nontargets) (*Figure 3—figure supplement 6*). We found that FMRP targets were significantly more likely to be directly bound by FMRP in cells as determined by CLIP than FMRP nontargets (*Figure 3D*; *Ascano et al., 2012*; *Darnell et al., 2011*; *Maurin et al., 2018*). We also found that FMRP targets had significantly higher densities of G-quadruplexes in their 3′ UTRs than FMRP nontargets, suggesting that this mode of localization target recognition may be conserved between mice and humans (*Figure 3E,F*).

## The RGG domain of FMRP specifically recognizes G-quadruplex sequences

Given our findings that FMRP localization targets were enriched for G-quadruplex sequences, we next sought to determine if FMRP directly interacted with G-quadruplex RNAs over other RNAs in vitro. To this end, we performed RNA Bind-n-Seq (RBNS), a quantitative and unbiased method that defines the specificity of a given RBP or RNA binding domain by incubating purified protein with a diverse pool of $10^{11}$ random RNA sequences (*Dominguez et al., 2018*; *Lambert et al., 2014*; *Taliaferro et al., 2016b*). In this technique, the input RNA pool and protein-associated RNA are sequenced using high-throughput sequencing. RNA sequences that mediate protein binding are identified as those enriched in the protein-bound RNA relative to the input pool (*Figure 4A*). Importantly, this assay uses a single round of RNA selection and uncovers a continuous range of RBP specificity.

We set out to determine which domains within FMRP, if any, contained the ability to specifically interact with G-quadruplex sequences. FMRP contains two domain types that are known to bind

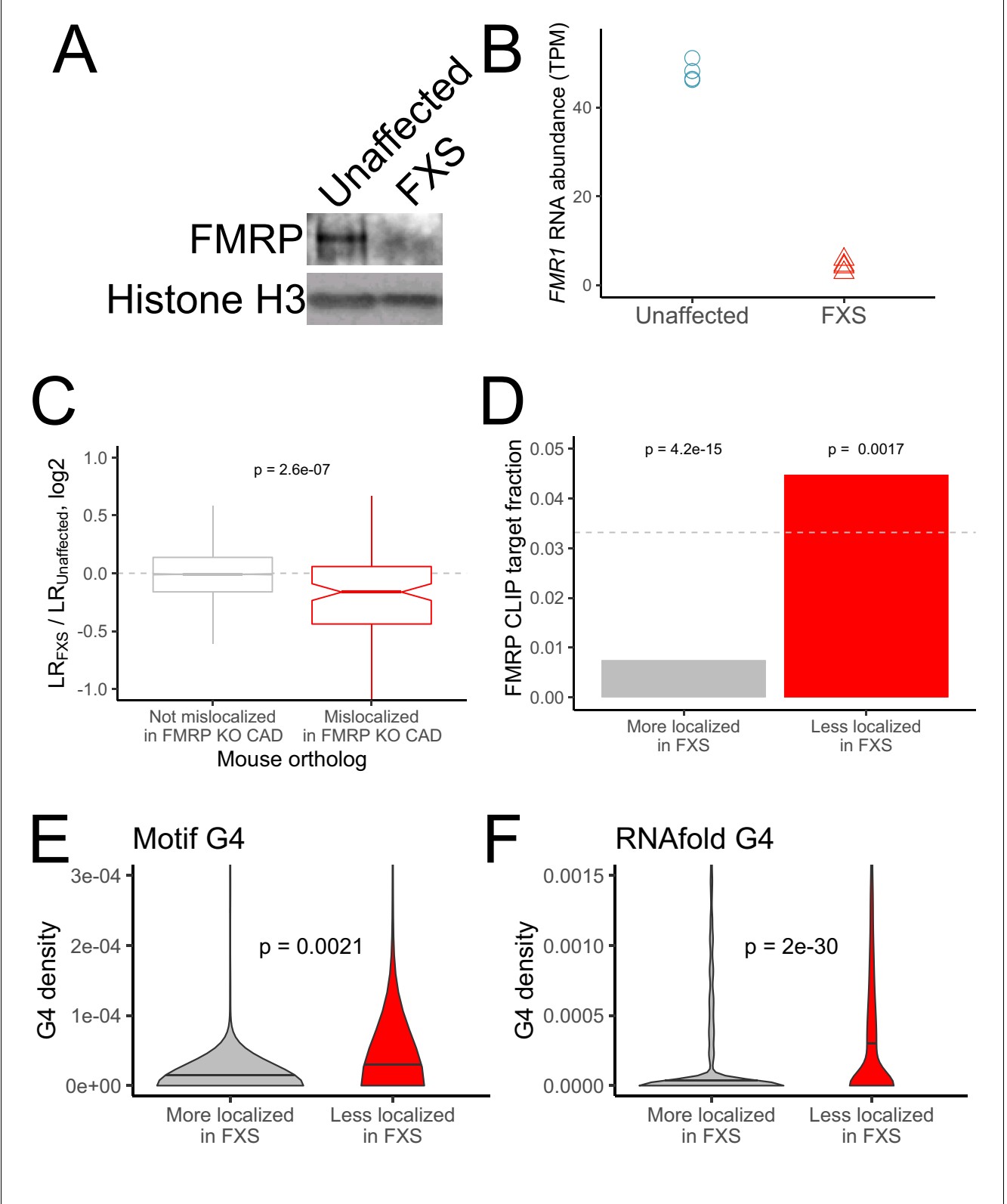

**Figure 3.** RNA mislocalization in FXS neurons. (**A**) FMRP protein expression in motor neurons differentiated from iPS cells derived from unaffected and FXS patients. (**B**) *FMR1* transcript abundance in motor neurons differentiated from iPS cells derived from unaffected and FXS patients. (**C**) Relative LR values comparing unaffected and FXS motor neurons. Human genes were binned based on whether or not their mouse orthologs were defined as FMRP localization targets (red) or nontargets (gray) in mouse CAD cells. (**D**) Fraction of human FMRP localization targets (red) and nontargets (gray), as

*Figure 3 continued on next page*

*Figure 3 continued*

defined in iPS-derived motor neurons, that were previously identified as bound by FMRP in cells (**Ascano et al., 2012**; **Darnell et al., 2011**). The null, expected fraction is represented by a gray dotted line (E-F) G-quadruplex densities in the 3′ UTRs of human FMRP localization targets (red) and nontargets (gray) as defined using a regular expression (E) and RNAfold (F). Y-axis values depict densities as in **Figure 2C and D**. P values were calculated using a Wilcoxon rank-sum test.

The online version of this article includes the following figure supplement(s) for figure 3:

**Figure supplement 1.** Schematic overview of motor neuron differentiation (top) and representative cell images (bottom).

**Figure supplement 2.** Protein dot blot assaying the efficiency of motor neuron fractionations.

**Figure supplement 3.** LR values for all genes (gray), ribosomal protein genes (red), and genes that are part of the electron transport chain (blue).

**Figure supplement 4.** PCA analysis of gene expression values from the soma and neurite compartments of wildtype and FXS motor neurons.

**Figure supplement 5.** Correlation between FMRP-dependent changes in LR values observed in mouse (FMRP-null CAD - wildtype CAD) and human (FXS motor neurons - unaffected motor neurons) neuronal cells.

**Figure supplement 6.** LR values for genes observed in unaffected and FXS motor neurons.

RNA, a pair of KH domains and a single RGG domain (**Figure 4A**). We purified recombinant fragments of mouse FMRP that included either the KH domains or the RGG domain (**Figure 4B**). We then performed RBNS with a pool of random 40mers in the presence of either potassium, which promotes G-quadruplex folding, or lithium, which inhibits G-quadruplex folding (**Hardin et al., 1992**). For each condition, we sequenced approximately 10 million RNA sequences in from both the input and protein-associated RNA pools. G-quadruplex RNA sequences, defined either as matches to specific motifs or through RNA folding predictions, were approximately 10-fold enriched in the RGG-associated RNA compared to the input pool (**Figure 4C,D**). Furthermore, the addition of lithium drastically reduced this enrichment of G-quadruplex sequences, indicating that the G-quadruplexes must be folded in order to interact with the RGG domain in vitro, as has been previously observed (**Menon et al., 2008**; **Ramos et al., 2003**; **Schaeffer et al., 2001**; **Zhang et al., 2014**).

Using fluorescence polarization assays, we found that the RGG domain interacted with a G-quadruplex RNA in the presence of potassium with a high affinity (Kd ~30 nM). This value is comparable to previous estimates of the affinity (**Zhang et al., 2014**). Repeating the assay in the presence of lithium showed reduced binding between the RGG domain and G-quadruplex RNA (**Figure 4—figure supplement 1**).

The KH domains of FMRP did not specifically interact with G-quadruplex sequences (**Figure 4C, D**), and we were unable to uncover any strong sequence specificity in the KH-bound RNA nor derive the affinity of the KH domain for any RNA. These data suggest that FMRP binds its RNA localization targets through interactions between its RGG domain and target transcripts.

## The RGG domain of FMRP as well as a G-quadruplex sequence in target mRNAs are both required for efficient localization of RNAs

To investigate a functional relationship between the RGG domain of FMRP and G-quadruplex sequences within 3′ UTRs of localized mRNAs, we used single molecule FISH (smFISH) to monitor the localization of a reporter transcript in CAD cells that contained either no FMRP, full length FMRP, or a truncation of FMRP lacking the RGG domain. These cells were made by rescuing an FMRP knockout line with GFP, FMRP, or FMRPΔRGG, respectively (**Figure 5A**). The reporter transcript contained the coding sequence of Firefly luciferase followed by the 3′ UTR of the Nol3 gene (Nol3 3′ UTR). This UTR was chosen because we observed in our sequencing data that Nol3 was mislocalized in FMRP knockout CAD cells (LR log2 fold change = −0.9, p=3e-14). In separate experiments, we also monitored the localization of a version of the reporter transcript in which the only RNAfold-predicted G-quadruplex sequence had been removed (Nol3-G4 3′ UTR) (**Lorenz et al., 2011**). The smFISH probes used to visualize the two reporters were identical and targeted the coding sequence of the transcript (**Figure 5A**).

Both the *FMR1* rescue constructs and the reporter transcripts were integrated into the genome of the CAD cells using Cre-lox recombination into a single genomic site (**Khandelia et al., 2011**). Using this strategy, cell pools were virtually homogeneous for integration of the rescue constructs, and the expression of both the rescue constructs and the reporter transcripts were controllable with doxycycline. Protein levels were assayed via western blot with or without doxycycline induction to confirm expression of rescue constructs. We found that the expression level of FMRP in the rescued

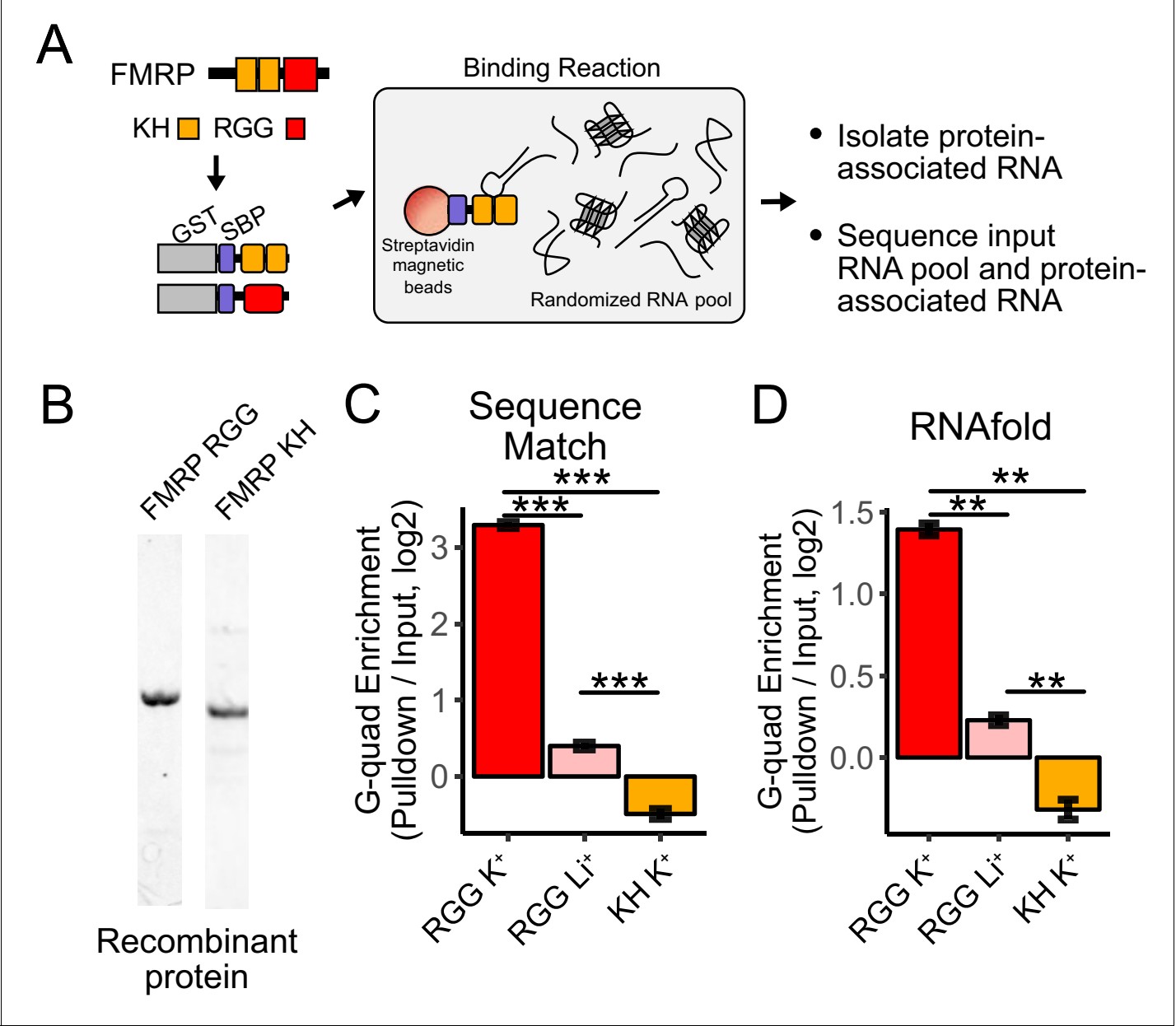

**Figure 4.** RNA Bind-n-Seq (RBNS) of FMRP domains. (**A**) Schematic representation of RBNS assay and domain architecture of FMRP. (**B**) Coomassie stain of purified recombinant FMRP fragments used for RBNS experiments. (**C**) Enrichments for G-quadruplex RNA sequence motifs in the protein-associated RNA relative to input RNA in the presence of potassium (red) and lithium (blue). Bars represent means of bootstrapped subsets. Error bars represent the standard deviation across subsets. (**D**) As in C, enrichments for G-quadruplex RNA sequences as predicted by computational RNA folding in protein-associated RNA relative to input RNA. Wilcoxon rank-sum p values < 0.01 are represented as (**), and p values less than 0.001 are represented as (***).

The online version of this article includes the following figure supplement(s) for figure 4:

**Figure supplement 1.** Fluorescence polarization values from FMRP RGG/RNA interaction assays.

cells was similar to the expression of FMRP in wildtype cells (***Figure 5B***). We then monitored the efficiency of reporter transcript localization to neurites by smFISH and quantified the results using FISH-quant (***Mueller et al., 2013***). We observed that the reporter transcript that contained a G-quadruplex sequence in its 3′ UTR was efficiently localized to neurites, but only in cells expressing full length FMRP (FMRP, Nol3 3′ UTR), and not in cells either completely lacking FMRP (GFP, Nol3 3′ UTR) or in cells expressing only a truncated form of FMRP (ΔRGG, Nol3 3′ UTR) (***Figure 5C,D***;

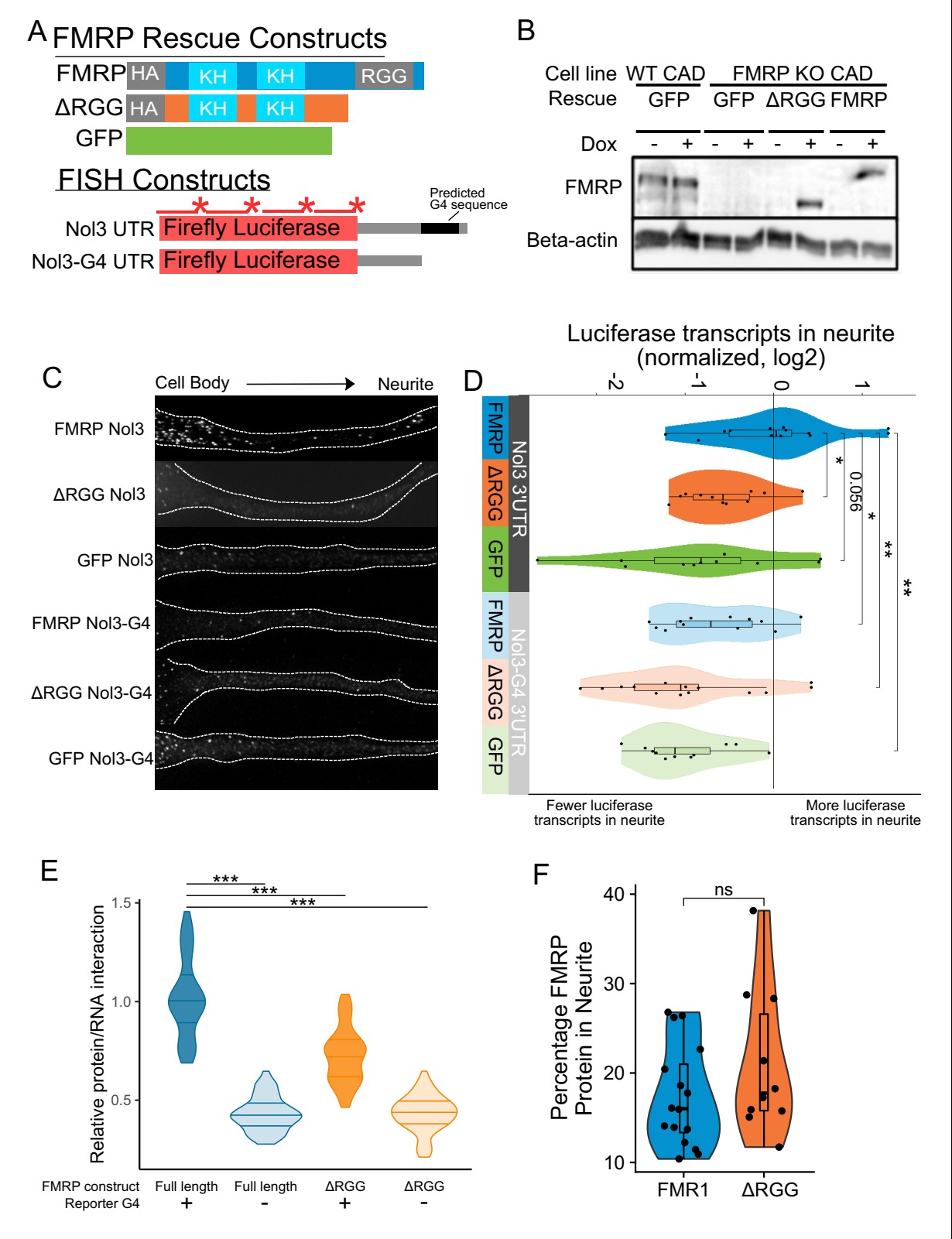

**Figure 5.** Efficient localization of a reporter transcript requires both a G-quadruplex sequence and the RGG domain of FMRP. (**A**) FMRP rescue constructs and smFISH reporter constructs used. smFISH probes, represented by red bars with asterisks, hybridize to the ORF of both reporter constructs. (**B**) Expression of FMRP rescue constructs in FMRP knockout-rescue CAD cells. (**C**) Representative Firefly luciferase transcript smFISH images. Images are oriented such that the cell body is toward the left. (**D**) Quantification of smFISH images. For all conditions, the number of reporter

*Figure 5 continued on next page*

*Figure 5 continued*

transcripts in the neurite was calculated using FISH-quant. Each dot in each condition represents a single cell. Transcript counts were normalized to those observed in the full length FMRP, Nol3 UTR condition. P values were calculated using a Wilcoxon rank-sum test. (E) Quantification of FMRP/Firefly luciferase RNA interaction using CLIP-qPCR. Wilcoxon rank-sum P values of less than 0.001 are represented by three asterisks. (F) Quantification of FMRP trafficking using immunofluorescence. For each cell, the proportion of fluorescence present in the neurite was quantified and normalized to total fluorescence in the entire cell. P values were calculated using a Wilcoxon rank-sum test.

The online version of this article includes the following figure supplement(s) for figure 5:

**Figure supplement 1.** Summary of smFISH results.

**Figure supplement 2.** Western blotting of IP samples from the CLIP-seq qPCR experiments.

*Figure 5—figure supplement 1*). Removal of the G-quadruplex sequence (Nol3-G4 3′ UTR) resulted in less efficient transcript localization regardless of the FMRP status of the cell (*Figure 5C,D*; *Figure 5—figure supplement 1*). These results are consistent with both the RGG domain of FMRP and a G-quadruplex sequence within the target transcript being required for efficient localization to neurites.

This observed defect in the transport of G-quadruplex-containing transcripts in the absence of the RGG domain of FMRP could be explained by two potential possibilities. It could be that the ΔRGG truncation was unable to bind the transcript in cells, or, alternatively, it could be that the ΔRGG truncation was unable to be transported to neurites. To distinguish between these possibilities, we tested the abilities of the ΔRGG truncation and the full length protein to bind the reporter transcripts in cells using CLIP-qPCR (*Yoon and Gorospe, 2016*) and assayed their ability to be transported to neurites using immunofluorescence.

To assay RNA binding ability, cells coexpressed one of the Firefly luciferase reporters, either containing or lacking a G-quadruplex (*Figure 5A*), and Renilla luciferase from a bidirectional promoter. RBPs were crosslinked to their bound RNAs using UV irradiation, and the rescuing FMRP construct was immunoprecipitated. By comparing the amounts of Firefly and Renilla luciferase transcripts in the input and immunoprecipitated samples using qPCR, relative levels of FMRP/Firefly RNA interaction in cells were determined. We found that the ΔRGG truncation showed significantly reduced interaction with the G-quadruplex-containing Firefly reporter compared to full length FMRP (*Figure 5E*, *Figure 5—figure supplement 2*). When we repeated the experiment using the Firefly reporter lacking the G-quadruplex, both full length FMRP and the ΔRGG truncation showed reduced, but importantly, equal, ability to bind the transcript (*Figure 5E*). These results are consistent with the in vitro results of the RBNS assay shown in *Figure 4*.

To address the ability of the ΔRGG truncation to be transported to neurites, we assayed the subcellular location of full length FMRP and the ΔRGG truncation using immunofluorescence. We found that cells expressing full length FMRP and the ΔRGG truncation displayed approximately equal proportions of the total cellular FMRP in their neurites, indicating that the full length protein and the truncation are trafficked to neurites at approximately similar levels (*Figure 5F*). Based on these results, we conclude that the observed defect in the localization of G-quadruplex-containing transcripts in cells expressing the ΔRGG truncation is likely due to a reduced ability of FMRP to bind the transcript and not a reduced ability of the protein to be trafficked.

To probe the functional relationship between the RGG domain of FMRP and G-quadruplex sequences transcriptome-wide, we fractionated the FMRP rescue lines into soma and neurite fractions and analyzed RNA from those fractions using high-throughput sequencing. We observed enrichments for mRNAs encoding ribosomal proteins and electron transport chain components in the neurites of all samples, indicating that fractionation of the cells was successful (*Figure 6—figure supplement 1*). Transcriptome-wide measures of RNA localization, as measured by LR values of all genes in the GFP and ΔRGG lines were strongly correlated. This is consistent with neither of these lines containing a localization-competent form of FMRP. Full length FMRP rescue samples were outliers in this analysis compared to the GFP and ΔRGG samples, perhaps due to the active localization regulatory activity of full length FMRP (*Figure 6A*, *Supplementary files 1c, 1d, 1e*).

We then compared changes in localization across samples by looking at changes in LR values for all genes. We found that comparing LR values in the full length FMRP rescue to either the GFP rescue or ΔRGG rescue produced highly concordant results (*Figure 6—figure supplement 2*), consistent with both comparisons involving one functionally active FMRP condition and one functionally

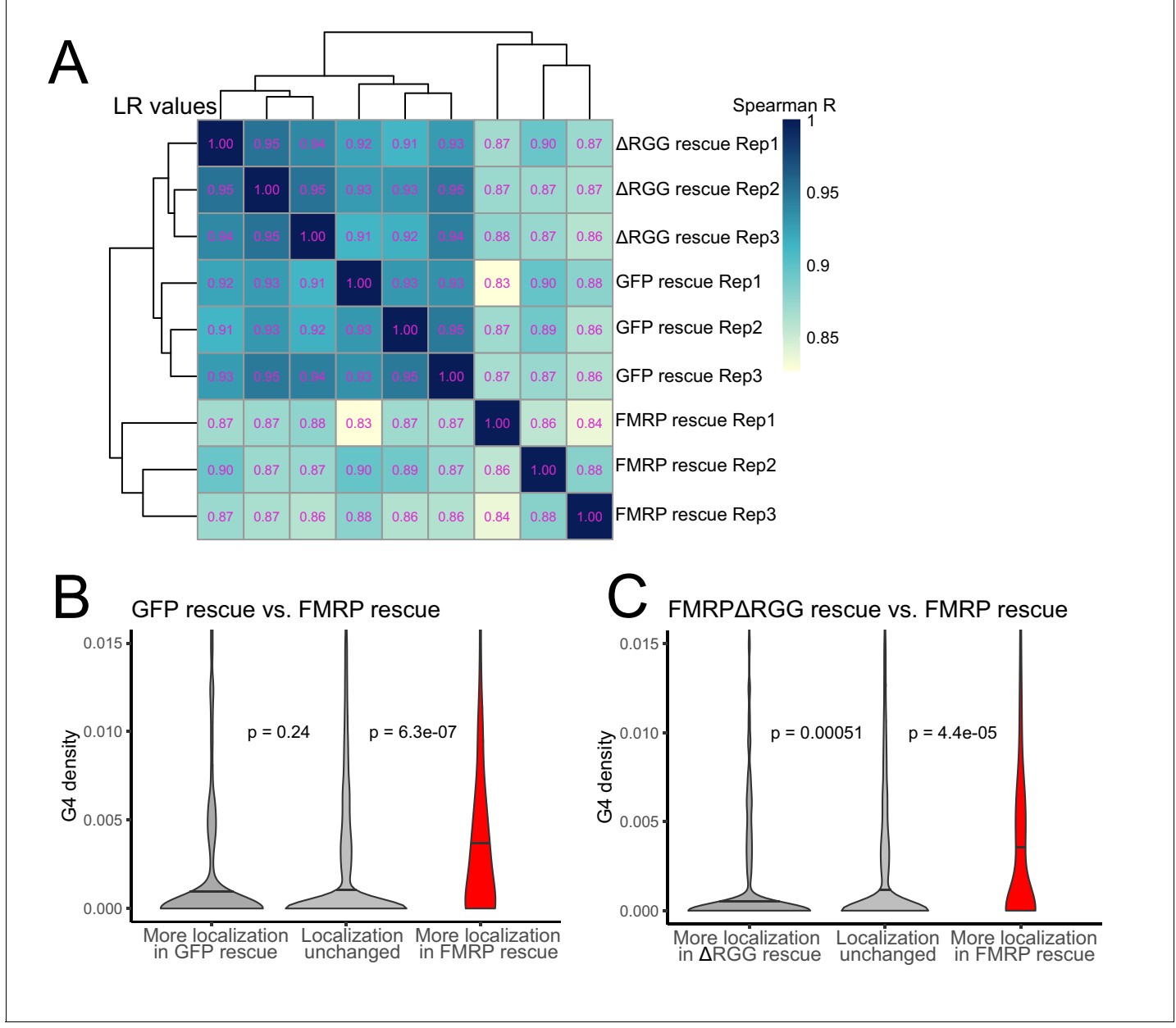

**Figure 6.** The RGG domain of FMRP is required for the localization of G-quadruplex-containing transcripts transcriptome-wide. (**A**) Correlation of LR values for all expressed genes in FMRP rescue samples. (**B**) G-quadruplex density, as measured by RNAfold, in the 3′ UTRs of transcripts that were more localized in the GFP rescue cells (dark gray, left) or full-length FMRP rescue cells (red, right). P values were calculated using Wilcoxon rank-sum tests. As in *Figure 2D*, values represent the number of guanosine residues participating in quadruplex sequences per nt. (**C**) As in (**B**), but comparing transcripts that were more localized in the FMRP RGG truncation rescue (dark gray, left) or the full length FMRP rescue (red, right).

The online version of this article includes the following figure supplement(s) for figure 6:

**Figure supplement 1.** LR values for all genes (gray), ribosomal protein genes (red), and genes that are part of the electron transport chain (blue).
**Figure supplement 2.** Correlation of delta LR values (nonfunctional FMRP condition [GFP, RGG, or KO] - functional FMRP condition [WT or FMRP]) between experiments.
**Figure supplement 3.** G-quadruplex density, as measured by RNAfold, in the 3′ UTRs of transcripts that were more localized in the GFP rescue cells (dark gray, left) or FMRP-RGG rescue cells (red, right).

inactive FMRP condition. These comparisons were also strongly positively correlated with our original CAD knockout experiment in which FMRP null and wildtype cells were compared (*Figure 6—figure supplement 2*).

Next, we compared the 3′ UTR G-quadruplex densities of genes that were differentially localized between given rescue conditions. When we compared the GFP rescue samples and full length FMRP rescue samples, we found that genes that were significantly more localized in the FMRP rescue cells contained significantly more G-quadruplex sequences in their 3′ UTRs than expected, consistent with our previous results (*Figures 6B* and *2D–F*). When we performed the same analysis comparing the FMRPΔRGG rescue samples to the full length FMRP rescue samples, we again found that genes whose localization was significantly more efficient in the full length FMRP samples contained significantly more 3′ UTR G-quadruplex sequences than expected (*Figure 6C*). This suggests that, transcriptome-wide, the RGG domain of FMRP is important for the neurite-localization of RNA molecules that contain G-quadruplex sequences. Consistent with this, we observed no relationship between the G-quadruplex content of an RNA and its differential localization in the GFP rescue and FMRPΔRGG rescue samples (*Figure 6—figure supplement 3*).

## The translational regulatory targets and RNA localization targets of FMRP are distinct pools of transcripts

FMRP is known to regulate mRNA metabolism at the level of RNA localization and translation (*Darnell et al., 2011*; *Dictenberg et al., 2008*). We wondered whether the transcripts targeted by these activities were largely the same or distinct. To address this question, we identified FMRP translational regulatory targets by performing ribosomal profiling on wildtype and FMRP null CAD cells (*Ingolia et al., 2009*). We observed that our ribosome protected fragments displayed strong enrichments for coding regions as well as a strong three nucleotide periodicity, indicating that the ribosome profiling data was of high quality (*Figure 7—figure supplements 1–4*).

We used the software package Xtail to identify transcripts that displayed significantly different ribosome occupancies in wildtype and FMRP null CAD cells (*Figure 7—figure supplement 5*, *Supplementary File 1f*; *Xiao et al., 2016*). Previous studies had reported that FMRP predominantly negatively regulates translation by inducing ribosome stalls (*Darnell et al., 2011*). Since stalls may result in ribosome pileups and increased ribosome densities, we reasoned that FMRP translation targets should have a lower ribosome occupancy in FMRP null cells, even though their rate of translation and protein output is increased in the absence of FMRP (*Figure 7A*). This is an important consideration, especially when investigating a protein that is believed to induce ribosome stalling. Cycloheximide-based ribosome profiling data informs on the locations of all ribosomes, not just translating ribosomes, and this is why we prefer the term 'ribosome occupancy' to 'translational efficiency' when referring to this data. We found genes that displayed reduced ribosome occupancy in FMRP null cells compared to wildtype cells were significantly more likely to have been identified as FMRP CLIP targets than expected (*Figure 7B*; *Ascano et al., 2012*; *Darnell et al., 2011*; *Maurin et al., 2018*), consistent with FMRP either generally acting to inhibit translation by inducing ribosome stalling or with FMRP generally acting as a translational activator (*Greenblatt and Spradling, 2018*).

We found a significant but weak correlation between the mislocalization (as measured by changes in LR) and translational repression (as measured by changes in ribosome occupancy) of transcripts due to FMRP (R = 0.08, p=4.1e-16). However, when we looked at the transcripts that were statistically significantly affected in the two conditions, we found no significant overlap between the translational regulatory targets and the RNA localization targets of FMRP, suggesting that the protein exerts these activities on largely distinct pools of transcripts (*Figure 7C*). While the localization targets of FMRP were enriched for genes that regulate transport in cells (*Figure 1—figure supplement 7*), the translational regulatory targets were enriched for genes that regulate RNA metabolism and for those that regulate synaptic activity (*Figure 7—figure supplement 6*).

We also found no relationship between the 3′ UTR G-quadruplex content of a transcript and whether or not it was translationally regulated by FMRP (*Figure 7D*). Identifying transcript features associated with translational regulation by FMRP proved difficult as the transcript features most strongly associated with FMRP translation targets were the GC content and length of the transcript's CDS (*Figure 7—figure supplements 7* and *8*). Translational regulatory targets of FMRP were strongly enriched for having GC-poor, long CDS sequences. These effects are difficult to

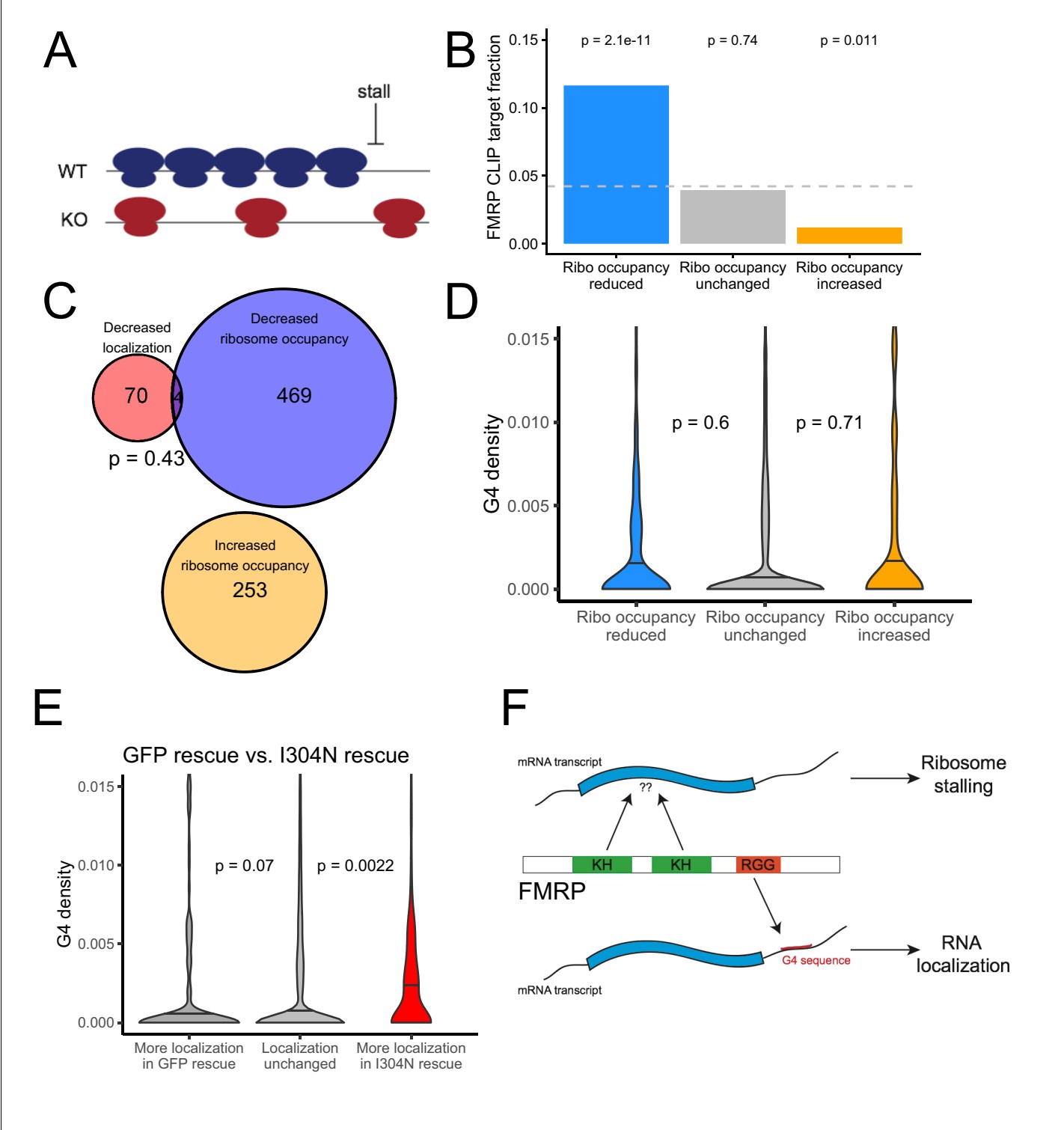

**Figure 7.** The translational regulation targets of FMRP are distinct from its RNA localization targets and have no relationship to G-quadruplex sequence density. (A) Schematic of how the removal of FMRP, which is known to cause ribosome stalls, may affect overall ribosome density on a transcript. (B) Fraction of genes whose ribosome occupancy significantly decreased (blue) or increased (orange) in the FMRP null samples that were previously identified as bound by FMRP in cells. The null expectation is represented by a gray dotted line. P values were calculated using a binomial test. (C) Overlap of FMRP localization targets (red) and translational repression targets (blue). P values were calculated using a Binomial test. (D) G-quadruplex density, as determined by RNAfold, for the 3' UTRs of transcripts whose ribosome occupancy was decreased (blue) or increased (orange) in the FMRP-null samples. P values were calculated using a Wilcoxon rank-sum test. (E) G-quadruplex density, as measured by RNAfold, in the 3' UTRs of transcripts

*Figure 7 continued on next page*

*Figure 7 continued*

that were more localized in GFP rescue cells (dark gray, left) or FMRP I304N rescue cells (red, right). P values were calculated using Wilcoxon rank-sum tests. (**F**) Model for how FMRP recognizes its targets. RNA localization targets are recognized through the interaction of the RGG domain and a G-quadruplex sequence in the 3′ UTR of the transcript. Translational regulation targets are recognized through the interaction of one or both KH domains with an unknown transcript feature, or perhaps through interaction with ribosomes.

The online version of this article includes the following figure supplement(s) for figure 7:

**Figure supplement 1.** Fraction of reads that map to the indicated genomic regions.

**Figure supplement 2.** Fraction of reads of that are the indicated lengths.

**Figure supplement 3.** Fraction of reads that map to the indicated reading frames.

**Figure supplement 4.** Metagene analysis of read densities across transcripts.

**Figure supplement 5.** Ribosome occupancy values, as calculated by Xtail, in wildtype and FMRP null CAD cells.

**Figure supplement 6.** Gene ontology analysis of the translational regulatory targets of FMRP in CAD cells.

**Figure supplement 7.** Metagene analysis of GC content across transcripts whose ribosome occupancy increases (blue) or decreases (red) in FMRP null CAD cells compared to wildtype cells.

**Figure supplement 8.** Effect sizes and significance of the relationships between transcript features and whether or not a transcript's ribosome occupancy was increased or decreased by FMRP.

**Figure supplement 9.** Quantification of FMRP trafficking using immunofluorescence.

**Figure supplement 10.** G-quadruplex density, as measured by RNAfold, in the 3′ UTRs of transcripts that were more localized in the I304N rescue cells (dark gray, left) or wildtype FMRP rescue cells (red, right).

**Figure supplement 11.** Expression of the different HA-tagged FMRP rescue constructs used in the FMRP null CAD line.

disentangle, however, as transcript GC content and length are correlated with each other in multiple species (*Marín et al., 2003*) as well as with a variety of other metrics, including conservation, expression, and RNA secondary structure.

Given that the target pools of the translation and RNA localization activities of FMRP appear to be distinct, we wondered whether these two activities could be functionally separated. To address this, we assayed the ability of a point mutant of FMRP, I304N, to promote localization of G-quadruplex-containing transcripts. The I304N mutation is in one of the KH domains of FMRP and renders the protein unable to interact with polysomes (*Feng et al., 1997*) and unable to repress translation (*Laggerbauer et al., 2001*). As with the ΔRGG truncation, we created CAD cells that express the I304N mutant by rescuing FMRP-null cells through site-specific integration of an I304N cDNA using Cre-lox recombination (*Khandelia et al., 2011*). We fractionated these cells into soma and neurite fractions, and then sequenced RNA collected from both fractions. Quality control metrics for the fractionation indicated that it was of high quality (*Figure 1—figure supplements 6*, *8* and *9*). We then compared RNA localization, transcriptome-wide, between FMRP-null cells rescued with GFP and cells rescued with I304N.

We found that transcripts that were more efficiently localized in the I304N sample contained significantly more G-quadruplexes than expected, indicating that the I304N mutant retains the ability to promote the localization of G-quadruplex-containing transcripts (*Figure 7E*, *Supplementary file 1g*). In support of this result, we found that wildtype FMRP and the I304N mutant were transported to neurites with approximately equal efficiencies, as had been previously reported (*Figure 7—figure supplement 9*; *Schrier et al., 2004*; *Wang et al., 2008*).

When we compared localization data from cells expressing full length, wildtype FMRP to those expressing the I304N mutant, we found that the mutant was deficient in promoting the localization of G-quadruplex-containing messages compared to the wildtype (*Figure 7—figure supplement 10*, *Supplementary file 1h*). However, the I304N mutant is approximately four fold more lowly expressed than the wildtype in these cells (*Figure 7—figure supplement 11*). This decreased expression of the I304N mutant is consistent with previous reports of reduced I304N expression in I304N mouse mutants (*Zang et al., 2009*). Given that in our system the wildtype and I304N mutant proteins are integrated and expressed from the same locus using the same promoter, we propose that this decrease in I304N protein expression is due to decreased protein stability.

From these results, we conclude that an FMRP mutant that is unable to regulate translation is still able to regulate the localization of G-quadruplex-containing transcripts. We propose a model in which the recognition of a G-quadruplex sequence within the 3′ UTR of a transcript by the RGG domain of FMRP is associated with localization of the transcript. Conversely, the recognition of

some as-yet unidentified transcript feature by one or both of the KH domains of FMRP may be associated with translational regulation (*Figure 7F*). This transcript feature, if it exists, has been difficult to identify as both bioinformatic sequence analysis of the translational regulation targets of FMRP and the RNA sequences specifically bound by the KH domains of FMRP in the RBNS experiment yielded no strongly enriched motifs.

This model is supported by three observations. First, although some studies have found that the RGG domain of FMRP is required for efficient interaction of the protein with polyribosomes (*Blackwell et al., 2010*), other studies have found that the RGG domain of FMRP is at least partially dispensable for this interaction while the KH domains are absolutely required (*Chen et al., 2014*; *Darnell et al., 2005b*; *Mazroui et al., 2003*). Second, the I304N mutation, which is in one of the KH domains, abolishes polyribosome association (*Feng et al., 1997*) and translation repression activity (*Laggerbauer et al., 2001*), but not localization activity. Third, G-quadruplex sequences are unable to compete with polysomes for binding to FMRP (*Darnell et al., 2005a*). Taken together, these findings support the idea that the dual functionalities of FMRP are driven by distinct binding capabilities of its different RNA binding domains.

## Discussion

Hundreds to thousands of transcripts are asymmetrically distributed within cells, yet the mechanisms that regulate trafficking are known for only a handful (*Cajigas et al., 2012*; *Gumy et al., 2011*; *Lécuyer et al., 2007*; *Taylor et al., 2009*; *Wang et al., 2012*). In this study we have used genome engineering, subcellular fractionation, and high-throughput sequencing to identify transcripts that depend on FMRP for transport in neuronal cells. These FMRP localization targets displayed a large enrichment for G-quadruplex sequences, particular in their 3′ UTRs, suggesting that FMRP specifically recognizes these sequences to target them for transport. G-quadruplex sequences were not enriched in the translational regulatory targets of FMRP as identified by ribosome profiling. In fact, no recognizable sequence motif was strongly enriched in the translational regulatory targets beyond a negative correlation with GC content. These apparent dual modes of target recognition and the lack of clear specificity in the translational regulatory targets may explain some of the previously reported conflicting results in regards to the preferred RNA binding sites of FMRP (*Anderson et al., 2016*).

Our results raise the question as to how an RNA that is targeted to neurites through the action of FMRP is not subject to translational regulation by FMRP. Although we did not test these hypotheses here, we propose these potential mechanisms. We found that binding events that lead to RNA transport frequently occur in the 3′ UTR, while other studies have found that binding events associated with translation regulation occur in the open reading frame (*Darnell et al., 2011*). Perhaps where FMRP contacts the RNA plays a role in the regulatory effects it can exert. Alternatively, the two activities of the protein could be differentially regulated spatially or in response to stimulus.

FMRP has long been known to bind to G-quadruplex sequences (*Anderson et al., 2016*; *Darnell et al., 2001*; *Suhl et al., 2014*), and G-quadruplex sequences have been shown to be sufficient for transport to neurites (*Subramanian et al., 2011*). In this work, we formally and directly combine those facts to functionally relate the ability of FMRP to bind G-quadruplexes with its ability to transport RNA molecules. It is important to note that we cannot formally prove that the G-quadruplex sequences enriched in FMRP localization targets are actually folded in cells. Recent reports have demonstrated that the majority of RNA sequences that have the ability to fold into G-quadruplex structures in vitro may be generally unfolded in cells (*Guo and Bartel, 2016*). Still, previous reports and our data demonstrate that the G-quadruplexes must be folded in order to be efficiently bound by FMRP (*Subramanian et al., 2011*), and G-quadruplex sequences have been reported to be under purifying selection (*Lee et al., 2020*).

No clear links have yet been found between the mislocalization of a specific transcript or transcripts in FXS neurons and FXS phenotypes. The relative contribution of the RNA localization and translational regulation activities of FMRP to FXS phenotypes similarly remain unclear. However, the fact that the I304N mutant, which is unable to inhibit translation but still able to regulate RNA localization, is associated with FXS phenotypes clearly points to the translational regulatory activity of FMRP being important. Further work is needed to more fully understand mechanistic links between RNA localization and neurological diseases including FXS.

For the vast majority of localized transcripts, in neurons and elsewhere, the RBP/RNA interactions that mediate transport are completely unknown. Classically, experiments designed to address these questions have focused on one transcript at a time and take the approach of systematic deletions and/or mutations to identify minimal sequence elements that regulate localization (*Ferrandon et al., 1994*; *Jambor et al., 2014*; *Kim-Ha et al., 1993*; *Kislauskis et al., 1994*). While these experiments have been successful and provided key insights for the field, they are quite labor-intensive, and it is difficult to derive general rules that govern RNA localization from isolated, single-transcript examples. RBP-perturbation experiments followed by transcriptomic profiling have been highly successful in elucidating mechanisms that underlie other RNA metabolic processes, including pre-mRNA processing, translation, and decay (*Iwasaki et al., 2016*; *Mukherjee et al., 2011*; *Wang et al., 2012*). By extending these techniques to the study of RNA localization, we envision this study as providing a framework for the investigation of other RBPs and their effects on RNA trafficking. Given that RNA mislocalization has been associated with a range of neurological diseases (*Wang et al., 2016*), these studies can lay groundwork that links molecular defects to observed phenotypes.

## Materials and methods

**Key resources table**

| Reagent type (species) or resource | Designation | Source or reference | Identifiers | Additional information |
|---|---|---|---|---|
| Gene (*Mus musculus*) | Fmr1 | | ENSMUS G00000000838 | |
| Sequence-based reagent | Fmr1 guide RNA | | | Used to create mouse FMR1-null cells, cloned into pX330 AAATTATCAGCTGGTAATTT |
| Cell line (*Mus musculus*) | CAD | Sigma | 08100805-1VL, RRID:CVCL_0199 | |
| Cell line (*Mus musculus*) | CAD/loxP | *Khandelia et al., 2011* | | Contains single integration of loxP cassette |
| Transfected construct (*Mus musculus*) | Fmr1 cDNA | Dharmacon | BC079671.1 | |
| Antibody | Mouse anti FMR1, monoclonal | Proteintech | 66548 | 1:5000 dilution for immunoblotting |
| Antibody | Mouse anti HA, monoclonal | Genscript | GenScript Cat# A01244, RRID:AB_1289306 | 1:2000 for immunofluorescence |
| Sequence-based reagent | smFISH probes against Firefly luciferase | BioSearch | VSMF-1006–5 | |
| Commercial assay or kit | Zymo Quick-RNA Microprep kit | Zymo Research | R1050 | |
| Other | Cell culture inserts for fractionatoin | Corning | 353102 | |

### Construction of an FMRP-null CAD line

CAD cell lines were authenticated using STR profiling and were found to be mycoplasma negative. The guide RNA sequence AAATTATCAGCTGGTAATTT was cloned into pX330 (gift of F. Zhang) using the BbsI sites. This plasmid was then transfected into CAD cells using Lipofectamine LTX. Forty-eight hours after transfection, the cells were sorted to single cells using a flow cytometer and allowed to grow without interference for two weeks. Clones were screened for FMRP expression using a polyclonal FMRP antibody (MBL International RN016P). Lesions at the locus of cutting were identified by PCR using genomic DNA to amplify the locus followed by Topo cloning of the PCR fragment and Sanger sequencing of individual bacterial colonies.

## Sequencing of subcellular transcriptomes from wildtype and FMRP null CAD cells

CAD cells were grown in DMEM/F12 (Gibco 11320033) supplemented with 10% FBS at 37C in 5% $CO_2$. To fractionate CAD cells into soma and neurite fractions, CAD cells were plated on porous, transwell membranes that had a pore size of 1.0 micron (Corning 353102). One million CAD cells were plated on each membrane. These membranes fit in one well of a six well plate, and six membranes were combined to comprise one single preparation. After allowing the cells to attach, the cells were induced to differentiate into a more neuronal state through the withdrawal of serum. Cells were then differentiated on the membranes for 48 hr.

To fractionate the cells, the membranes were washed one time with PBS. One mL of PBS was added to the top (soma) side of the membrane. The top side of the membrane was then scraped repeatedly and thoroughly with a cell scraper and the soma were removed into an ice-cold 15 mL falcon tube. After scraping, the membranes were removed from their plastic housing with a razor blade and soaked in 500 µL RNA lysis buffer (Zymo R1050) at room temperature for 15 min to make a neurite lysate. During this time, the 6 mL of soma suspension in PBS was spun down and resuspended in 600 µL PBS. 100 µL of this soma sample was then carried forward for RNA isolation using the Zymo QuickRNA MicroPrep kit (Zymo R1050). In parallel, RNA was also isolated from the 500 µL neurite lysate using the same kit. Typically, the neurite RNA yield from the six combined membranes was 500–1000 ng. The efficiency of the fractionation was assayed by Western blotting using anti-beta-actin (Sigma A5441, 1:5000) and anti-Histone H3 (Abcam ab10799, 1:10000) antibodies.

PolyA-selected, stranded libraries were constructed using 25 ng total RNA and an Illumina Neoprep library preparation system. The resulting libraries were then sequenced on an Illumina NextSeq 500 sequencer using paired-end, 75 bp reads. Approximately 40 million read pairs were obtained per sample. Each condition (wildtype and FMRP null; soma and neurite) was prepared and sequenced in triplicate for a total of 12 samples.

## Quantification of RNA localization from CAD subcellular sequencing data

Transcript abundances were calculated from read files using kallisto (CAD Fmr1 knockout) (*Bray et al., 2016*) or using salmon v0.11.1 (CAD Fmr1 rescue samples, FXS iPS samples) (*Patro et al., 2017*) with the options –seqBias and –gcBias. Transcript abundances were then collapsed to gene abundances using txImport (*Soneson et al., 2015*). LR values for genes were calculated as log2 of the ratio of neurite/soma normalized counts for a gene produced by DESeq2 using the salmon/tximport calculated counts (*Love et al., 2014*). Genes with less than 10 counts in any sample were excluded from further analyses. To identify genes whose LR values significantly changed across conditions, the Xtail software package was used (*Xiao et al., 2016*). Xtail is designed to process ribosome profiling data and identify genes whose ribosome occupancy changes across conditions. However, since ribosome occupancy is simply a ratio of expression count values (ribosome footprints/RNAseq), we reasoned that since LR is also a ratio of expression count values (neurite/soma), Xtail could be used to identify genes whose LR value changed across conditions. For further analyses of this data, mislocalized genes were defined as those with a FDR of less than 0.01.

## Identification of transcript features associated with mislocalized genes

To ask if previously identified FMRP-interacting transcripts were enriched in our set of mislocalized genes, we defined FMRP-interacting transcripts as the intersection of RNA targets identified in two earlier studies (*Ascano et al., 2012*; *Darnell et al., 2011*) as was previously done (*Suhl et al., 2014*).

To identify 6mer sequences enriched in the 3′ UTRs of mislocalized genes, we calculated the abundance of all 6mers in the UTRs of mislocalized genes and, as a control, in the UTRs of expressed but non-mislocalized genes. These abundances were then compared using a Fishers exact test, and the resulting p values were corrected for multiple hypothesis testing using the Benjamini-Hochsberg method.

To computationally identify predicted G-quadruplex sequences, we used two approaches. First, to identify WGGA sequences separated by variable length linkers, we searched sequences using the regular expression (?=([AU]GGA(.{0,6})[AU]GGA(.{0,6})[AU]GGA(.{0,6})[AU]GGA)). Alternatively, we analyzed sequences using a sliding window approach and RNAfold (*Lorenz et al., 2011*). The

window was 80 nt wide and slid 20 nt at a time. The sequence within each window was analyzed using RNAfold and the -g option. Nucleotides with a '+' for their predicted structure status were designated as nucleotides participating in a quadruplex.

For analyses involving experimentally defined G-quadruplex sequences, data was taken from previously published experiments using RNA purified from mouse embryonic stem cells that utilized RT stops that occurred with differential frequency in G-quadruplex promoting (potassium) conditions and G-quadruplex inhibiting (lithium) conditions (*Guo and Bartel, 2016*). The data taken from this paper contained RT stop sites, the 60 nt upstream of the stop site, and the frequency at which those sites were observed in the potassium and lithium conditions. RT stops representing G-quadruplexes were defined as those which were observed at least five times more frequently in the potassium data than in the lithium data. For those sites, the G-quadruplex region was defined as the 60 nt immediately upstream of the stop.

Cliff's delta values representing the effect sizes associated with each transcript feature and its association with FMRP-mediated localization were calculating using the orddom package in R (*Rogmann, 2013*).

## G-quadruplex conservation analyses

To ask if G-quadruplexes were positionally conserved, PhyloP scores were calculated for all G nucleotides in all 3' UTRs. If the G nucleotides were predicted by RNAfold to be participating in quadruplexes as discussed above, their scores were put in one bin. If they were not predicted to be participating in quadruplexes, they were put in another bin. The scores of the G nucleotides in the two bins were then compared.

FXS iPSC line generation and iPSC culture iPSCs were generated by transducing control and FXS patient fibroblasts with Sendai virus from the Cytotune 2.0 Reprogramming Kit (Life Technologies) per the manufacturer's protocol. Briefly, early passage fibroblasts were cultured in fibroblast medium (10% ES-qualified FBS, 0.1 mM NEAA, 55 µM β-mercaptoethanol, high glucose DMEM with Glutamax) for two days. On day 0, fibroblasts were transduced with Sendai virus encoding KOS, hc-Myc, hKlf4, each at an MOI of 5. Cells were fed with fibroblast medium every other day for one week. On day 7, cells were passaged onto vitronectin (Life Technologies) coated dishes at a density of 250,000 to 500,000 cells/well. Beginning on day 8, cells were fed every day in Essential eight medium (Life Technologies) and colonies began to emerge within another 7–10 days. Individual iPSC colonies were manually picked and transferred to a dish coated with either vitronectin or Matrigel (BD) and expanded as clonal lines. All iPSC lines were karyotyped (WiCell) and characterized for expression of markers of pluripotency using immunofluorescence and RT-PCR. Additionally, we confirmed loss-of-function of *FMR1* in our FXS patient iPSC lines by taqman assays for *FMR1* mRNA and western blotting for FMRP. iPSCs were maintained on Matrigel coated dishes and fed every day with complete mTesR1 medium (Stem Cell Technologies). iPSCs were passaged every 5–7 days using ReLeSR (Stem Cell Technologies).

## FXS iPS line differentiation into motor neurons

Motor neurons were generated using an improved, suspension culture-based protocol based on previously published Dual-Smad inhibition approaches (*Chambers et al., 2009*; *Du et al., 2015*; *Reinhardt et al., 2013*). iPSCs were grown as clusters in suspension, and differentiated in a set-wise fashion as outlined in Supp. *Figure 3a*. Briefly, clusters were exposed to Smad and ROCK inhibition and WNT activation during the first 6 days, followed by activation of the retinoic and sonic hedgehog pathways. At day 10 the clusters were dissociated, and single cells were plated on membranes coated with Matrigel (Corning) in maturation media until d18-21 when MNs were analyzed.

## Fractionation of iPS neurons into soma and neurite fractions

Generally, iPS neurons were fractionated in a very similar method as the CAD cells. However, since the iPS neurons usually contained less material, instead of combining six wells together into one prep as was done for the CAD cells, between 6 and 12 wells were combined together. After mechanical fractionation by scraping into soma and neurite fractions and RNA isolation using the Zymo Quick RNA Microprep Kit, between 100 and 500 ng of neurite RNA was obtained.

Fractionation efficiency was monitored by protein dot blot using the beta-actin and histone H3 antibodies described above. PolyA-selected, stranded RNAseq libraries were constructed using the Kapa mRNA Hyperprep kit, 100 ng of total RNA, and 15 PCR cycles of library amplification. Each condition (FXS and unaffected; soma and neurite) was fractionated, prepared, and sequenced with four replicates for a total of 16 samples. Samples were sequenced using paired-end, 150 nt sequencing on an Illumina NovaSeq. Approximately 30 million read pairs were obtained per sample.

## Analysis of iPS neuron subcellular transcriptome data

Raw sequencing data from the iPS soma and neurite samples were analyzed using salmon and tximport as described above. After analysis of changes in RNA localization between FXS and unaffected samples using Xtail, very few genes met a reasonable FDR threshold for being significantly different between conditions. This may be a reflection of the increased variance in expression values in these samples relative to the CAD samples, perhaps due to slight variations between preps in the differentiation from iPS cells into motor neurons. For the analysis of G-quadruplex densities in mislocalized genes, mislocalized genes were identified as those who had a log2 fold change in LR across conditions (FXS vs unaffected) of at least 0.25.

## RNA Bind-n-Seq protein purification and sequence analysis

Rosetta *E. coli* cells were transformed with pGEX6 bacterial expression constructs (GE Healthcare) harboring GST-SBP (streptavidin binding peptide)-RGG or the two KH domains from mouse FMRP. The RGG domain was defined as residues 478–614 and the KH domains were defined as residues 222–315 from mouse FMRP1. Cultures were grown in LB broth until the optical density reached ~0.8. Cultures were brought to 16℃ and induced with IPTG overnight. Cells were then harvested and lysed in lysis buffer (25 mM Tris-HCl, 150 mM NaCl, 3 mM MgCl2, 1% Triton X-100, 500units/per 1L culture Benzonase Nuclease, 20 mg/L Lysozyme, Pierce Protease Inhibitor tablets). Cell pellets were then sonicated, incubated for 30 min on ice, and centrifuged at 12500 rpm for 40 min. Supernatants were passed through a 0.45 micron filter and GST-tagged proteins were purified using GST-trap FF columns (GE). Samples were further subjected to size exclusion chromatography (Superdex 200 Increase 10/300 GL column (GE)). Pure fractions were pooled together and concentrated by centrifugation (Amicon Ultra-0.5mL Centrifugal Filter Units). Purity was assessed by PAGE and Coomassie stain.

The RNA Bind-n-Seq assay was performed similar to previously described (*Dominguez et al., 2018*). Briefly, RNA with a random region of 40nt was prepared by in vitro transcription. The RNA pool resolved by PAGE and purified. Concentrated RNA was heated at 85C for 5 min in the presence of 150 mM KCl or LiCl and allowed to cool for at least 3 min to allow for folding. Binding reactions were performed as previously described (*Dominguez et al., 2018*). All wash and binding buffers contained either 150 mM KCl or LiCl. Protein-RNA complexes were eluted, RNA isolated and prepared for sequencing as previously described (*Dominguez et al., 2018*).

Sequence enrichments in RNA Bind-n-Seq data were obtained in three ways. First, kmer frequencies (k = 3–6) were computed in protein-bound libraries and compared to kmers in the input library. For each kmer, the ratio of its frequency in the protein-bound library to the input library results in an enrichment or 'R' value for every kmer, as was previously reported (*Dominguez et al., 2018*). To identify G quadruplex sequences, we determined the fraction of protein-bound reads with matches to a variety of G quadruplex motifs (e.g. GGGN1-7GGGN1-7 GGGN1-7GGG) and compared that to the number of input reads with the same motif. Lastly, we used RNAfold version 2.4.11 (*Lorenz et al., 2011*) to predict G quadruplexes (RNAfold -g option) in the 40 nucleotide randomized regions of input and bound RNAs. As with the pattern match approach, the ratio of protein-bound to input reads predicted to fold into G quadruplexes was computed. In each case reads were subsampled for estimates. Error bars shown in figures represent standard deviation from mean across subsamples.

## Fluorescence polarization assay

RNAs labeled at the 3′ end with 6-FAM were purchased from Integrated DNA Technologies. The RNA sequences used were: 5′-GGGAGGGAGGGAGGGUA-6FAM-3′ (rG4) and 5′-CCCCCC UCCCCCCU-6FAM-3′ (CTRL RNA). rG4 was heated to 100℃ in the presence of 150 mM KCl

(promoting G-quadruplex formation) or 150 mM LiCl (disrupting G-quadruplex formation) for 3 min and allowed to cool down to room temperature for at least 15 min. rG4 was then mixed with binding buffer containing 150 mM KCl (or LiCl), 10 mM Tris-HCl and 5% glycerol. Cation usage (K+ or Li+) was kept constant throughout the assay. The RGG domain of FMRP was purified and then serially diluted with binding buffer to concentrations between 1 µM to 2 nM. The RGG domain was then incubated with 10 nM of rG4 in potassium, rG4 in lithium or CTRL RNA. After incubation for 5 min at 25°C, plates were centrifuged at 1000 rpm for 1 min. Fluorescence polarization was monitored with a PHERAstar plate reader (BMG Labtech) at 26–27°C.

## Construction of an FMRP-null CAD line with single LoxP cassette

CAD LoxP cell lines were received from the lab of Eugene Makeyev (*Khandelia et al., 2011*). This line contains a single loxP cassette in the genome of the cells to facilitate recombination of exogenous constructs. A knockout insertion cassette was introduced to exon 6 of the endogenous *Fmr1* gene via CRISPR-Cas9 technology to generate CAD LoxP FMRP KO cells. This cassette includes stop codons in all frames as well as a strong polyadenylation signal. 300 nt of homology on either side of the gRNA cut site was included to facilitate homology directed repair. A G418 resistance gene was included to select for integration. pX330 was used to express Cas9 and the guide RNA sequence AAATTATCAGCTGGTAATTT. Following selection with G418, cells were sorted to single cells using a flow cytometer and allowed to expand for two weeks. Clones were screened for insertion of the cassette by PCR as well as FMRP mRNA and protein depletion determined by qPCR, RNA-seq and Western Blot Analysis with a monoclonal FMRP antibody (Proteintech 66548). Lesions at the locus of cutting were identified by PCR using genomic DNA to amplify the locus followed by Topo cloning of the PCR fragment and Sanger sequencing of individual bacterial colonies.

## Expression of Fmr1 rescue constructs in CAD cells

CAD LoxP FMRP KO cell lines were plated in 12-well plates at 1.0 to 1.5 × 10^5 cells per well in an DMEM:F12 supplemented with 10% Equafetal 12 to 18 hr before transfection. Cells were then co-transfected with a derivative of pRD-RIPE (containing one of three *Fmr1* rescue cDNAs and Firefly Luciferase fused with one of two 3'UTRs) blended with 1% (wt/wt) of a Cre-encoding plasmid (pCAGGS-Cre). *Fmr1* cDNAs were derived from cDNA clone BC079671.1. To transfect one well of a 12-well plate, 0.5 µg of purified plasmid was mixed with 2 µL Lipofectamine LTX reagent, 1 µL PLUS reagent and 50 µL Opti-MEM following the manufacturer's protocol. Cells were incubated with the transfection mixtures overnight, the medium was changed and the incubation continued for another 24 hr before addition of puromycin (5 µg/ml). Incubation with puromycin continued for several days until untransfected control wells were depleted of cells. The cultures with stably integrated constructs were expanded in a full growth medium. Lysates were collected following 2 µg/mL doxycycline induction to assay construct expression via western blot.

## Immunoblot of FMRP rescue constructs

Expression of rescue constructs in the CAD FMRP KO cell lines were determined by Western blot analysis. Cells with integrated rescue constructs were plated in 6-well plates at 2–3 × 10^5 cells per well in duplicate. Each cell line was grown in full growth media or full growth media supplemented with 2 µg/mL doxycycline for 48 hr. Lysates were collected with 200 µL RIPA buffer, incubated on ice for 15 min then passed through a 20G needle 20 times. Lysates were stored at −80°C. Denatured lysates were separated by PAGE on 4–12% Bis-Tris gradient gels (Invitrogen) along with Spectra Multicolor Broad Range (Fisher Scientific) using MOPS SDS NuPAGE Running Buffer (Invitrogen) and NuPAGE LDS Sample Buffer (Invitrogen) on ice at 200 V for 1 hr. Gels were transferred to PVDF membranes (Millipore) using NuPAGE Transfer Buffer (Invitrogen) at 15 V for 55 min before being washed three times, 5 min each with PBST at room temperature (RT) with agitation and blocked with 5% Milk Powder (Sigma) in PBST for 1 hr at RT with agitation. Blots were again washed three times, 5 min each in PBST at RT with agitation before being incubated in 1:5000 FMRP mouse monoclonal (Proteintech) primary antibody in 2% Milk Powder in PBST overnight at 4C with agitation or with 1:10,000 Histone H3 mouse monoclonal antibody (Abcam) in 2% milk powder in PBST for 2 hr at room temperature with agitation. Blots were washed three times, 5 min each at RT with agitation and then transferred to Anti-mouse IgG HRP-conjugated secondary antibody in 2% Milk Powder in

PBST and incubated at RT for 2 hr. Blots were washed again as previously described and visualized using WesternBright HRP Substrate Kit (Advansta).

## smFISH of reporter transcripts

CAD LoxP FMRP KO cells with Integrated Rescue constructs were plated on PDL coated glass coverslips (neuVitro) within 12-well plates at approximately $2.5 \times 10^4$ cells per well in full growth media. Cells were allowed to attach for 2 hr before changing to serum depleted media supplemented with 2 µg/mL doxycycline. Cells were allowed to express constructs and differentiate for 48 hr. Cells were washed once with PBS before being fixed in 3.7% formaldehyde in PBS for 10 min at room temperature then washed twice with PBS before permeabilization with 70% Ethanol at 4°C for 6–8 hr. Cells are incubated in smFISH Wash Buffer at room temperature for 2–5 min. Per hybridization reaction, 2 µL of Stellaris FISH Probes, Firefly Luc with Quasar 670 Dye were added to 200 µL of Hybridization Buffer. An empty opaque tip box is prepared with wet paper towels below with parafilm covering the top. Hybridization buffer with probes (200 µL per reaction) was 'beaded' on top of the parafilm and the PDL coated glass coverslips were placed cell side down onto 'beaded' buffer. The entire tip box is incubated at 37°C over night. Glass coverslips were transferred to fresh 12-well plates cell side up and incubated with smFISH Wash Buffer at 37°C in the dark for 30 min. Buffer was then replaced with fresh smFISH Wash Buffer supplemented with 100 ng/mL DAPI and incubated in the dark at 37°C. DAPI stain was replaced with fresh smFISH Wash Buffer and incubated for 5 min at room temperature. Coverslips were then mounted onto slides with 6 µL Fluoromount G and sealed with nail polish. Slides were imaged on a widefield DeltaVision Microscope with consistent laser intensity and exposure times across samples.

## smFISH quantification

Experimental parameters within FISH-quant were set to match widefield DeltaVision microscope and the Stellaris FISH probes (*Tsanov et al., 2016*). The excitation and emission wavelengths used were 547 nm and 583 nm respectively which were recorded by a 60X objective with 1.4 numerical aperture with 1.33 refractive index oil. This was captured by a camera such that pixel sizes in the xy were 98.4 nm wide and z pixels a 398 nm wide. Cell outlines were drawn using cell autofluorescence in the FITC channel where somas and neurites were treated as separate cells. Images were first filtered with a 3D 2x gaussian method using the default parameters. Pre-detection was conducted, and spots were fit using FISH-quant default settings and a pixel-intensity threshold set to overestimate detected spots. All thresholding parameters were set tightly around each mean population to exclude gross outliers. Averaged detected spots were inspected for spherical character to ensure quality of detection. These detection settings were the same for every soma and neurite imaged.

## Immunofluorescence for HA tagged FMRP rescues in differentiated neurons

CAD FMRP KO cells with doxycycline inducible HA-tagged rescue constructs were plated on top of PDL coated glass coverslips (neuVitro) within 12-well plates at approximately $2.5 \times 10^4$ cells per well in in full growth media supplemented with 2.0 µg/mL doxycycline for 2 hr to allow attachment. Media was removed and replaced with differentiation media (lacking serum) supplemented with 2.0 µg/mL doxycycline for an additional 48 hr to induce neurite growth and rescue construct expression. Media was removed and cells were washed with PBS before being fixed with 3.7% formaldehyde at room temperature for 10 min. Cells were washed twice with PBS following fixation for 5 min each. Cells were permeabilized for 30 min at room temperature with 0.3% Triton-X in PBS. Primary HA antibody (GenScript A01244-100) was diluted 1:2000 in PBS containing 1% BSA and 0.3% Triton-X for 1 hr at 37°C. Cells were then rinsed three times with PBS before adding 100 ng/mL DAPI and fluorescent secondary antibody (Cell Signaling Technology 4413S) diluted 1:500 in PBS containing 1% BSA and 0.3% Triton-X for 1 hr at room temperature. Cells were again rinsed three times with PBS then mounted onto slides with Fluoromount G and sealed with nail polish. Slides were imaged on a widefield DeltaVision Microscope with consistent laser intensity and exposure times across samples.

## CLIP-qPCR

One 15 cm plate of cells was used per condition. Expression of the HA-tagged FMRP constructs, Firefly luciferase, and Renilla luciferase were induced with doxycycline for 48 hr prior to the start of the experiment. To crosslink, cells were washed once with cold PBS, then irradiated with 400 mJ / cm$^2$ of 254 nm ultraviolet light in a Stratlinker while on ice. Cells were then scraped in 5 mL PBS, centrifuged for 5 min at 2000 g at 4°C, and resuspended in 400 µL lysis buffer (20 mM Tris pH 7.5, 100 mM KCl, 5 mM MgCl$_2$, 0.5% NP-40, 1 mM DTT, 1 mM PMSF). 3 µL of RNase inhibitor were added, and the lysate was incubated on ice for 10 min, then spun at 10,000 g for 15 min at 4°C.

Anti-HA magnetic beads (Pierce) were prepared by washing them three times with buffer A (50 mM Tris pH 7.5, 150 mM NaCl, 0.05% Tween). Lysate was then added to the beads and incubated at 4°C with rotation for 3 hr. Beads were then washed three times with buffer A and incubated with 10U DNase (ThermoFisher) in buffer A supplemented with the 1X buffer that accompanied the enzyme. Beads were then incubated with 20 µg proteinase K (ThermoFisher) in proteinase K buffer (20 mM Tris pH 7.5, 0.1% SDS, 100 mM NaCl) at 55°C for 15 min. The supernatant was then removed from the beads and RNA was isolated using an RNA Clean and Concentrator kit (Zymo).

Firefly luciferase and Renilla luciferase transcript levels were monitored in input and IP samples using Taqman qPCR with probe sets supplied by Life Technologies. Firefly luciferase RNA was quantified using a VIC-labeled probe while Renilla luciferase RNA was quantified using a FAM-labeled probe. The reaction was performed using Taqman Fast Advanced Master Mix (Life Technologies).

## Ribosome footprinting of CAD cells

CAD cells were grown in the absence of serum for 48 hr to induce neuronal differentiation. Ribosome footprint libraries and companion RNAseq libraries were then produced using the Truseq Ribo Profiling Kit from Illumina according to the manufacturer's instructions. Libraries were sequenced on an Illumina NextSeq sequencer, with about 40 million reads being obtained per sample. Three techical replicates were performed per condition (WT and knockout; ribosome protected fragments (RPF) and RNAseq) for a total of 12 samples.

Gene level abundance estimates from the RPF and RNAseq data were performed using kallisto and tximport as described above (*Bray et al., 2016*). Ribosome occupancy values and their changes between wildtype and knockout cells were then calculated using Xtail (*Xiao et al., 2016*).

## Acknowledgements

We thank Christopher Burge for initial support with this study and members of the Taliaferro lab for helpful discussion.

This work was funded by the RNA Bioscience Initiative at the University of Colorado Anschutz Medical Campus (JMT), a Webb-Waring Early Career Investigator Award from the Boettcher Foundation (AWD-182937) (JMT), NIH1R35GM133885 (JMT), NIHDK120444 (HAR), NIHAI140044 (HAR), NIHDK118803-01A1 (LIH) NIH1R01MH109026 and 1U54HD082013 (GJB). It was further supported by a Pre-doctoral Training Grant in Molecular Biology (NIH-T32-GM008730) (RG), a Guild ACORN Postdoctoral Fellowship (LIH), and grants from the Colorado Clinical and Translational Science Institute (HAR), the Children's Diabetes Foundation (HAR), a new investigator award from the NIH/HIRN consortium (HAR), the Culshaw Junior Investigator Award (HAR), and the Juvenile Diabetes Research Foundation (HAR).

## Additional information

### Funding

| Funder | Grant reference number | Author |
|---|---|---|
| Boettcher Foundation | Webb-Waring Early Career Investigator Award AWD-182937 | J Matthew Taliaferro |
| National Institutes of Health | NIH1R35GM133885 | J Matthew Taliaferro |
| National Institutes of Health | NIHDK120444 | Holger A Russ |

| National Institutes of Health | NIHAI140044 | Holger A Russ |
|---|---|---|
| National Institutes of Health | NIHDK118803-01A1 | Laura I Hudish |
| National Institutes of Health | NIH1R01MH109026 | Gary Bassell |
| National Institutes of Health | 1U54HD082013 | Gary Bassell |
| National Institutes of Health | NIH-T32-GM008730 | Raeann Goering |
| Guild ACORN | | Laura I Hudish |
| Colorado Clinical and Translational Sciences Institute | | Holger A Russ |
| Children's Diabetes Foundation | | Holger A Russ |
| NIH | NIH/HIRN consortium new investigator award | Holger A Russ |
| Culshaw Junior Investigator Award | | Holger A Russ |
| Juvenile Diabetes Research Foundation | | Holger A Russ |
| Human Islet Research Network | NIH/HIRN consortium new investigator award | Holger A Russ |
| University of Colorado | RNA Bioscience Initiative | J Matthew Taliaferro |

The funders had no role in study design, data collection and interpretation, or the decision to submit the work for publication.

## Author contributions

Raeann Goering, Formal analysis, Validation, Investigation, Visualization, Methodology, Writing - original draft, Writing - review and editing; Laura I Hudish, Resources, Data curation, Formal analysis, Investigation, Methodology; Bryan B Guzman, Data curation, Formal analysis, Investigation, Writing - review and editing; Nisha Raj, Resources, Validation, Investigation, Methodology; Gary J Bassell, Holger A Russ, Resources, Supervision, Funding acquisition, Methodology, Writing - review and editing; Daniel Dominguez, Resources, Supervision, Investigation, Methodology, Writing - review and editing; J Matthew Taliaferro, Conceptualization, Formal analysis, Supervision, Funding acquisition, Investigation, Visualization, Methodology, Writing - original draft, Project administration, Writing - review and editing

## Author ORCIDs

Raeann Goering (iD) https://orcid.org/0000-0002-0351-335X
Bryan B Guzman (iD) https://orcid.org/0000-0002-2711-9533
Nisha Raj (iD) https://orcid.org/0000-0002-5980-5001
Holger A Russ (iD) https://orcid.org/0000-0001-5117-2927
J Matthew Taliaferro (iD) https://orcid.org/0000-0001-7580-1433

## Decision letter and Author response

Decision letter https://doi.org/10.7554/eLife.52621.sa1
Author response https://doi.org/10.7554/eLife.52621.sa2

# Additional files

## Supplementary files

• Supplementary file 1. Xtail outputs for differential localization or ribosome occupancy of transcripts between two different conditions. (a) Xtail output for the differential localization of transcripts in wildtype and FMRP null CAD cells. All log2 fold change values are knockout/wildtype. (b) Xtail output for the differential localization of transcripts in unaffected and FXS motor neurons. All log2 fold change values are FXS/unaffected. (c) Xtail output for the differential localization of transcripts in

FMRP null CAD cells rescued with either GFP or full length FMRP. (d) Xtail output for the differential localization of transcripts in FMRP null CAD cells rescued with either FMRP-RGG or full length FMRP. (e) Xtail output for the differential localization of transcripts in FMRP null CAD cells rescued with either FMRP-RGG or GFP. (f) Xtail output for the differential ribosome occupancy of genes in wildtype and FMRP null CAD cells. (g) Xtail output for the differential localization of transcripts in FMRP null CAD cells rescued with either GFP or I304N FMRP. (h) Xtail output for the differential localization of transcripts in FMRP null CAD cells rescued with either I304N or wildtype FMRP.

- Transparent reporting form

## Data availability

Raw sequencing data and processed files are available through the Gene Expression Omnibus, accession GSE137878.

The following dataset was generated:

| Author(s) | Year | Dataset title | Dataset URL | Database and Identifier |
|---|---|---|---|---|
| Goering R, Hudish LI, Russ HA, Taliaferro JM | 2020 | Regulation of RNA localization by FMR1 | https://www.ncbi.nlm.nih.gov/geo/query/acc.cgi?acc=GSE137878 | NCBI Gene Expression Omnibus, GSE137878 |

The following previously published datasets were used:

| Author(s) | Year | Dataset title | Dataset URL | Database and Identifier |
|---|---|---|---|---|
| Taliaferro JM, Vidaki M, Oliveira R, Olson S, Zhan L, Saxena T, Wang ET, Graveley BR, Gertler FB, Swanson MS, Burge CB | 2016 | Profiling of soma and neurite transcriptomes | https://www.ncbi.nlm.nih.gov/geo/query/acc.cgi?acc=GSE67828 | NCBI Gene Expression Omnibus, GSE67828 |
| Farris S, Ward JM, Carstens KE, Samadi M, Wang Y, Dudek SM | 2019 | Hippocampal Subregions Express Distinct Dendritic Transcriptomes that Reveal Differences in Mitochondrial Function in CA2 [RNA-seq] | https://www.ncbi.nlm.nih.gov/geo/query/acc.cgi?acc=GSE116342 | NCBI Gene Expression Omnibus, GSE116342 |
| Minis A, Dahary D, Manor O, Leshkowitz D, Pilpel Y, Yaron A | 2013 | Sub-Cellular Transcriptomics – Dissection of the mRNA composition in the axonal compartment of sensory neurons | https://www.ncbi.nlm.nih.gov/geo/query/acc.cgi?acc=GSE51572 | NCBI Gene Expression Omnibus, GSE51572 |
| Zappulo A, van den Bruck D, Mattioli C, Franke V, Imami K, McShane E, Moreno-Estelles M, Calviello L, Filipchyk A, Peguero-Sanchez E, Muller T, Woehler A, Birchmeier C, Merino E, Rajewsky N, Ohler U, Mazzoni EO, Selbach M, Akalin A, Chekulaeva M | 2017 | RNA localization is a key determinant of neurite-enriched proteome - RNAseq | https://www.ebi.ac.uk/arrayexpress/experiments/E-MTAB-4978/ | ArrayExpress, E-MTAB-4978 |

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
