## [Decision Letter]

Thank you for submitting your article "FMRP drives RNA localization to neuronal projections through interactions between its RGG box and G-quadruplex sequences" for consideration by *eLife*. Your article has been reviewed by three peer reviewers, including Gene W Yeo as the Reviewing Editor and Reviewer #1, and the evaluation has been overseen by James Manley as the Senior Editor. The following individual involved in review of your submission has agreed to reveal their identity: Stephanie Ceman (Reviewer #3).

The reviewers have discussed the reviews with one another and the Reviewing Editor has drafted this decision to help you prepare a revised submission.

Summary:

This manuscript evaluates the role of FMRP on mRNA localization to neurites using genomic approaches to reveal sequence elements and related protein domains that are suggested to be required for binding and/or neurite-targeting. The results mostly confirm current models of RGG-domain function in interacting with G-rich sequences for FMRP, and also confirm known FMRP's role in localizing neuritic mRNAs. Nevertheless, if the authors' argument for separation of function for RNA binding and its requirement for neurite-targeting, versus translation control by FMRP can be strengthened, that would be a novel finding and a requirement for publication.

Essential revisions:

1) Far stronger evidence that the neurite purification led to expected, enriched mRNAs in the neurites is required. e.g., using neurofilament markers to provide confidence that indeed neurites are being enriched in the purification. Comparing the neurite-enriched mRNAs in the induced neurons to published lists (Zappulo et al., 2017) will strengthen the manuscript.

2) Ascano et al. identified FLAG tagged FMRP targets in human non-neuronal, proliferative cells (HEK293?), and Darnell et al., 2011, had an additional step to isolate mouse brain polyribosomal mRNAs. This would imply that Ascano's FMRP targets are unlikely to be localized mRNAs and Darnell's targets are enriched for ribosome-bound mRNAs. Are these datasets really comparable? What is the number of intersecting targets? And are the intersected targets really what should be compared to find FMRP-dependent localized mRNAs? If I understand Figure 2A correctly, 15% of 86 genes (localization reduced) is 12 (seems like a relatively low number to rely on for the downstream non-parametric statistical tests). What would the overlap be if the individual CLIP datasets were used to compute overlaps, and there are other datasets for FMRP CLIPs available now – what are the overlaps there and how robust are the statistical outcomes downstream?

3) The reviewers appreciate the use of FXS neurons and do agree that relying on this dataset is preferable than the CAD neurons. Thus, it is important to ensure that the antibodies for FMRP is not cross-reactive to properly interpret the data. If the authors want to utilize the CAD system, it is necessary to evaluate for clarity if the same mRNAs are found localized in both human and mouse systems. Please provide the list of mRNAs identified to the readers.

4) Further clarity and evidence that the absence of the Nol3 reporter RNA in neurites in the delta RGG FMRP case is due to loss of RNA binding or loss or neurite-targeting. Is the RGG domain required for binding the 3'UTR/the RNA in the soma and is unable to localize to the neurites, or that it binds but cannot target. Do they also observe an effect on endogenous Nol3 localization? Can CLIP be performed to clarify binding to the Nol3 reporter/endogenous RNA in the presence/absence of the different domains? Truly decoupling localization and translation with luciferase or other assays would be strong.

5) Can the affinity of the RGG and KH domains for the pools of selected RNA in the RBNS assay be measured? Are there constant regions that flank the variable 40 nts? Could that contain a sequence that binds the KH domains?

---

## [Author Response]

Essential revisions:1) Far stronger evidence that the neurite purification led to expected, enriched mRNAs in the neurites is required. e.g., using neurofilament markers to provide confidence that indeed neurites are being enriched in the purification. Comparing the neurite-enriched mRNAs in the induced neurons to published lists (Zappulo et al., 2017) will strengthen the manuscript.

We presented evidence in Figure 1C and Figure 3—figure supplement 2 that the fractionations were clean. If there is leakage of the soma compartment into neurite samples, we would expect to see Histone H3 signal in the neurite samples. We have tried, both in the experiments for this publication and previous ones (Taliaferro et al., 2016), to detect neurofilament markers by Western blotting in our neurite samples but have not had success, likely due to the small amount of material collected from neurites.

As an alternative, we therefore chose to compare transcriptome-wide RNA localization values (projection / soma) in the data presented in this manuscript to those obtained in similar, previously published experiments (Figure 1—figure supplement 6). Overall, we compared projection / soma expression ratios from 65 samples spanning 2 species, 4 laboratories, and 9 cell types. We found that our fractionation data correlated quite well (R ~ 0.6 to 0.8) with almost all of these samples, and that our samples were generally indistinguishable from almost all other previously published ones. Keep in mind that we are correlating ratios of expression values, not just the expression values themselves.

To further analyze this, we found that the similarity of these samples to each was far greater than would be expected by chance (p = 2e-65, Figure 1—figure supplement 7), demonstrating that our data is quite consistent with previously published datasets.

We and many others have found that RNAs encoding ribosomal proteins and electron transport chain proteins are routinely enriched in projections. This was true of all of the samples presented in this manuscript, and the large majority of the 65 samples we analyzed (Figure 1—figure supplements 8 and 9), again demonstrating that our fractionations are consistent with previously published ones.

2) Ascano et al. identified FLAG tagged FMRP targets in human non-neuronal, proliferative cells (HEK293?), and Darnell et al., 2011, had an additional step to isolate mouse brain polyribosomal mRNAs. This would imply that Ascano's FMRP targets are unlikely to be localized mRNAs and Darnell's targets are enriched for ribosome-bound mRNAs. Are these datasets really comparable? What is the number of intersecting targets? And are the intersected targets really what should be compared to find FMRP-dependent localized mRNAs? If I understand Figure 2A correctly, 15% of 86 genes (localization reduced) is 12 (seems like a relatively low number to rely on for the downstream non-parametric statistical tests). What would the overlap be if the individual CLIP datasets were used to compute overlaps, and there are other datasets for FMRP CLIPs available now – what are the overlaps there and how robust are the statistical outcomes downstream?

These are important subtleties in these datasets that we are glad the reviewers pointed out. Using data from another publication that performed FMRP CLIP and compiled data from the Ascano and Darnell papers (Maurin et al., 2018) there were 7800 identified targets in the Ascano data and 768 identified targets in the Darnell data. There is an overlap of 597 targets between them, which is a significant overlap (p < 2e-16, binomial test).

We redid the analysis of overlaps in FMRP binding targets and our identified FMRP localization targets. We used the targets identified in each CLIP-seq dataset (Ascano, Darnell, and Maurin) individually to assess overlap with our list of functional targets. In each case, we observed a significant overlap (binomial test) between the CLIP targets and our localization targets. These plots are included as Figure 2—figure supplements 1, 3, and 5.

Additionally, we reoriented the analysis to ask about the localization sensitivity to FMRP loss for each of the identified sets of FMRP CLIP targets. If our analysis revealed direct targets of FMRP, we would expect that FMRP CLIP targets would be preferentially less localized in response to FMRP loss compared to FMRP CLIP nontargets. This is exactly what we found for each CLIP dataset. CLIP targets were significantly less localized following FMRP loss than expected. These results are presented in Figure 2—figure supplements 2, 4, and 6.

The results of these analyses, both the gene set overlap and localization change of CLIP targets in response to FMRP loss are also included in Figure 2A and B. The set of FMRP CLIP targets for these panels are defined as those shared in common between the Ascano, Darnell, and Maurin datasets.

In short, the results are robust, no matter the CLIP dataset used.

3) The reviewers appreciate the use of FXS neurons and do agree that relying on this dataset is preferable than the CAD neurons. Thus, it is important to ensure that the antibodies for FMRP is not cross-reactive to properly interpret the data. If the authors want to utilize the CAD system, it is necessary to evaluate for clarity if the same mRNAs are found localized in both human and mouse systems. Please provide the list of mRNAs identified to the readers.

We are slightly unclear on what the reviewers mean regarding cross-reactivity. If this is in regard to the Western blot demonstrating that FMRP expression is decreased in FXS samples (Figure 3A), this result is corroborated by the accompanying RNAseq data. In that data, *FMR1* transcript levels are decreased 10-20 fold in the FXS samples relative to unaffected controls (Figure 3B). Transcript levels of the paralogs FXR1 and FXR2 are unaffected. Given that the Western blot shows a clear decrease in protein expression, we believe that the most parsimonious interpretation of this data is that the antibody recognizes FMRP. Further, the FMRP antibody we used shows a clear and complete loss of signal in the CAD knockout line (Figure 1B and Figure 7—figure supplement 9), which would not be the case if there was cross-reactivity.

There is a strong correlation in localization data obtained from CAD cells and localization data obtained from the iPS neurons (R ~ 0.7). This is reflected in Figure 1—figure supplement 6 and has been now mentioned explicitly in the text. Localization data for all genes in the CAD system and the iPS system are presented in Supplementary file 1A and B, respectively.

4) Further clarity and evidence that the absence of the Nol3 reporter RNA in neurites in the delta RGG FMRP case is due to loss of RNA binding or loss or neurite-targeting. Is the RGG domain required for binding the 3'UTR/the RNA in the soma and is unable to localize to the neurites, or that it binds but cannot target. Do they also observe an effect on endogenous Nol3 localization? Can CLIP be performed to clarify binding to the Nol3 reporter/endogenous RNA in the presence/absence of the different domains? Truly decoupling localization and translation with luciferase or other assays would be strong.

These are important points. To address them, we performed two experiments.

In the first, we assayed the ability of full length FMRP and the RGG truncation to bind the Nol3 reporter in cells, either with or without the G-quadruplex. This was done using a CLIP-qPCR experiment. The Nol3 UTR is attached to the open reading frame of Firefly luciferase. The cells also coexpress *Renilla* luciferase from the same promoter that drives Firefly. By measuring the ratio of Firefly/*Renilla* transcripts using qPCR in Input and crosslink/IP samples, we can therefore measure the relative affinity that different FMRP constructs have for different UTRs.

We found that, in cells, relative to full length FMRP, the RGG truncation mutant was significantly reduced in its ability to interact with G-quadruplex-containing transcripts. Conversely, the ability of the RGG truncation to interact with transcripts that do not have a G-quadruplex was similar to that of the full length protein. These results are presented in Figure 5E and are consistent with our in vitro binding data presented in Figure 4.

To address the ability of the RGG truncation to be trafficked to neurites, we quantified the amount of FMRP protein in the soma and neurite using immunofluorescence. We found that full length FMRP and the RGG truncations displayed similar proportions of FMRP protein in the neurite, suggesting that trafficking of the RGG truncation is not significantly impaired. These results are presented in Figure 5F.

Based on these results, we conclude that it is likely the reduced ability of the RGG truncation to interact with G-quadruplexes that is responsible for the majority of the observed localization defects.

5) Can the affinity of the RGG and KH domains for the pools of selected RNA in the RBNS assay be measured? Are there constant regions that flank the variable 40 nts? Could that contain a sequence that binds the KH domains?

To address this, we performed fluorescence polarization assays to determine affinities. We found that the interaction between the RGG domain and a G-quadruplex RNA had an affinity of approximately 30 nM. This is in line with previous estimates of this affinity (60 nM, Zhang et al., 2014). We observed that lithium acted as a noncompetitive inhibitor in this assay, lowering the amount of RNA that was bound, but not changing the affinity. These data are presented in Figure 4—figure supplement 1.

We attempted to detect binding of the KH domains to RNA, either to the constant regions that flank the variable 40 nt or to the G-quadruplex RNA. We were never able to detect binding of the KH domains to either RNA sequence.